# Loc²: Interpretable Cross-View Localization via Depth-Lifted Local Feature Matching

**Zimin Xia**\*, **Chenghao Xu**\* & **Alexandre Alahi**
École Polytechnique Fédérale de Lausanne (EPFL), Switzerland
`{zimin.xia, chenghao.xu, alexandre.alahi}@epfl.ch`
\*Equal contribution `https://github.com/vita-epfl/Loc2`

## Abstract

We propose an accurate and interpretable fine-grained cross-view localization method that estimates the 3 Degrees of Freedom (DoF) pose of a ground-level image by matching its local features with a reference aerial image. Unlike prior approaches that rely on global descriptors or bird's-eye-view (BEV) transformations, our method directly learns ground–aerial image-plane correspondences using weak supervision from camera poses. The matched ground points are lifted into BEV space with monocular depth predictions, and scale-aware Procrustes alignment is then applied to estimate camera rotation, translation, and optionally the scale between relative depth and the aerial metric space. This formulation is lightweight, end-to-end trainable, and requires no pixel-level annotations. Experiments show state-of-the-art accuracy in challenging scenarios such as cross-area testing and unknown orientation. Furthermore, our method offers strong interpretability: correspondence quality directly reflects localization accuracy and enables outlier rejection via RANSAC, while overlaying the re-scaled ground layout on the aerial image provides an intuitive visual cue of localization performance.

## 1 Introduction

Visual localization, a fundamental task in computer vision and mobile robotics, aims to estimate the camera pose with respect to a representation of the environment (Thrun, 2002). In dense urban areas, specialized positioning sensors, such as Global Navigation Satellite System (GNSS), have errors up to tens of meters (Xia et al., 2021). Fine-grained cross-view localization has recently emerged as a promising complement. It estimates the 3 Degrees of Freedom (DoF) camera pose, i.e., 2D planar location and yaw orientation, by comparing captured ground-level images to an aerial image of the surroundings, identified using coarse GNSS measurements.

The key to cross-view localization is to establish associations between ground-level and aerial images. However, the extreme visual differences between the two views make it challenging for recent advances in image matching (Leroy et al., 2024; Sun et al., 2021; Wang et al., 2025) to produce reliable correspondences, and there is no ground-aerial pixel-level ground truth to finetune these methods. In practice, the majority of cross-view localization approaches seek global alignment between two views by matching global image descriptors (Xia et al., 2023) or aligning ground-view transformed bird's-eye-view (BEV) features with aerial images (Fervers et al., 2023b; Wang et al., 2024b). However, these methods offer limited interpretability, as they cannot explicitly identify which objects in the ground view correspond to those in the aerial view.

Recently, FG² (Xia & Alahi, 2025) demonstrated the feasibility of establishing local feature correspondences between ground and aerial images by propagating BEV point matches to the image plane. However, performing the matching in BEV is suboptimal, since warping a ground image into BEV inevitably introduces ray-directional distortions (Song et al., 2024) and discards information along the height dimension (Xia & Alahi, 2025; Wang et al., 2024a). This loss of information hinders matching to aerial images and, as a result, degrades localization performance, especially in challenging settings such as unknown camera orientation.

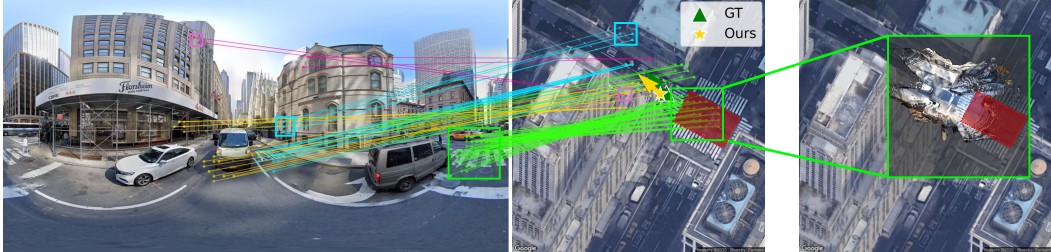

Figure 1: Loc$^2$: Interpretable cross-view **loc**alization via **loc**al feature matching. Loc$^2$ establishes accurate correspondences between aerial and ground views, with colors indicating distinct correspondence regions. Using the estimated rotation, translation, and scale, the ground view is warped onto the aerial image, providing a visual interpretation of localization quality.

Therefore, we propose to establish correspondences directly between ground and aerial images. Instead of relying on pixel-level annotations (Leroy et al., 2024; Wang et al., 2025), our approach is weakly supervised using only 3-DoF camera poses. To infer the pose from image-plane correspondences, we lift the sampled ground points into the BEV space with the help of monocular depth models (Yang et al., 2024a;b; Piccinelli et al., 2025; Wang & Liu, 2024). In unseen environments, usually only the non-metric relative depth prediction, i.e., depth up to an unknown scale, is reliable. Therefore, our method supports both metric and relative depth, and jointly estimates the camera pose and depth scale using scale-aware Procrustes alignment (Umeyama, 1991).

As shown in Fig. 1, our method offers superior interpretability. Since the pose is computed analytically from the matches, the quality of local feature correspondences directly reflects localization accuracy, and we find that the number of inlier correspondences shows a strong correlation with localization accuracy. Moreover, by overlaying the scaled, rotated, and translated ground BEV layout onto aerial images, our method provides an additional visual cue of localization accuracy.

Concretely, our main contributions are: (1) We propose a simple and accurate fine-grained cross-view localization method that matches local features across views. Our method achieves superior localization accuracy under challenging conditions such as cross-area generalization and unknown orientation. (2) Our method is highly interpretable. The pose is computed analytically from the estimated correspondences, which also enable outlier filtering via RANSAC. When relative depth is used at inference, our method estimates its scale, and the alignment between the re-scaled ground layout and the aerial image further provides a visual cue of localization quality. (3) By leveraging differentiable scale-aware Procrustes alignment, our local feature matching becomes end-to-end trainable using only camera pose supervision, without requiring pixel-level annotations.

## 2 RELATED WORK

**Fine-grained cross-view localization** is challenging due to the drastic viewpoint differences and asynchronous capture times between ground and aerial imagery. State-of-the-art approaches tackle this domain gap primarily by learning a deep network for camera pose estimation. Global descriptor-based methods (Xia et al., 2022; Lentsch et al., 2023; Xia et al., 2023) leverage contrastive learning to pull the ground and aerial features to a common representation space, and compare the ground image descriptor to aerial descriptors extracted at different candidate poses. Geometry transformation-based methods (Fervers et al., 2023b; Wang et al., 2023; Sarlin et al., 2024; Wang et al., 2024a; Song et al., 2024; Shi et al., 2023; Wang et al., 2024b; Fervers et al., 2023a; Xia & Alahi, 2025) first warp the ground image into a BEV representation, and then either perform a sliding-window search over the aerial image to find the best alignment (Sarlin et al., 2024; Fervers et al., 2023b; Shi et al., 2023) or estimate the relative pose by matching features between the BEV and aerial view (Song et al., 2024; Wang et al., 2024b; 2023; 2024a; Xia & Alahi, 2025). Despite the increasing localization accuracy, most methods do not explicitly identify matched local features across views, offering limited interpretability. A recent work (Xia & Alahi, 2025) demonstrated, for the first time, the ability to find ground-aerial local feature correspondences by defining a dense 3D point cloud and querying image features for each point. However, this process is inefficient due to the sparse nature of 3D space, and this method performs poorly when camera orientation is unknown.

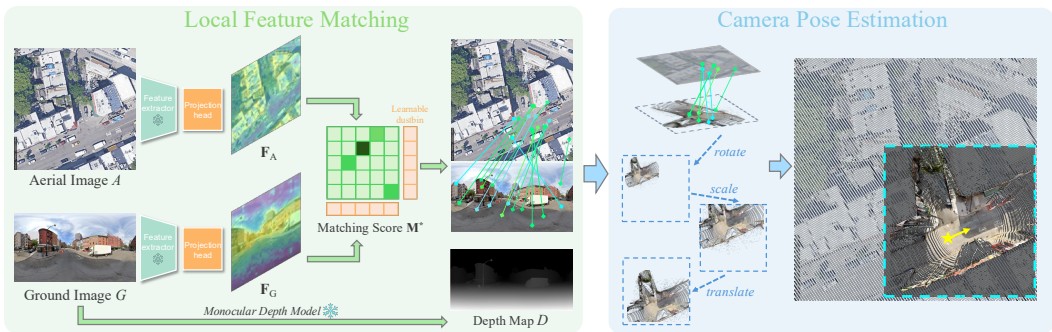

Figure 2: Overview of our proposed method. Our method first matches local features between ground and aerial images. The matched ground points are then lifted to the BEV space using monocular depth priors. By aligning these correspondences using scale-aware Procrustes alignment, we estimate the rotation, translation, and scale between the ground and aerial views.

**Ground-view visual localization** typically follows a highly interpretable pipeline (Sarlin et al., 2019; Sattler et al., 2016), which involves finding local feature correspondences (Sarlin et al., 2020; Sun et al., 2021) and then estimating camera pose using a geometry-based solver (Lepetit et al., 2009). However, this pipeline becomes challenging in the cross-view setting due to the significant domain gap and the lack of labeled correspondences. We propose to learn such correspondences for cross-view localization using only camera pose supervision, facilitated by monocular depth priors.

## 3 METHODOLOGY

Fine-grained cross-view localization estimates the 3-DoF pose, i.e., the 2D metric location $\mathbf{t} \in \mathbb{R}^2$ and the yaw orientation $o \in [-\pi, \pi)$, of a ground-level image $G$ by matching it to a geo-referenced aerial image $A$ covering the local area. As shown in Fig. 2, our method first establishes correspondences between $G$ and $A$ (Sec. 3.1), then lifts the matched ground points to the BEV using the predicted depth map $D = \mathcal{D}(G)$ from off-the-shelf monocular depth models, and finally computes the 3-DoF pose via scale-aware Procrustes alignment (Sec. 3.2). This fully differentiable pipeline enables end-to-end learning of image-plane matching under weak supervision from camera poses.

### 3.1 LOCAL FEATURE MATCHING

We estimate the 3-DoF metric camera pose by learning image-plane local feature correspondences between ground and aerial images. Our model has two feature extraction branches that share the same architecture. Each branch consists of a frozen DINOv2 feature extractor (Oquab et al., 2024) followed by our lightweight projection head, which is composed of several convolutional layers and a self-attention layer (Vaswani et al., 2017). The two branches map the aerial image $A$ and the ground image $G$ to feature maps $\mathbf{F}_A$ and $\mathbf{F}_G$, respectively.

Afterwards, we compute the pairwise matching scores $M$ between $\mathbf{F}_A$ and $\mathbf{F}_G$ using cosine similarity, $M = \text{cosine}(\mathbf{F}_A, \mathbf{F}_G)/\tau$, where $\tau$ is a temperature parameter. Following DeTone et al. (2018); Sarlin et al. (2020), we append a learnable dustbin to both the rows and columns of $M$, allowing the model to reject uncertain or unmatched points. On this extended matching score matrix $M^*$, we apply dual softmax normalization over rows and columns to obtain a match probability matrix $\hat{M}^*$,

$$\hat{M}^*_{ij} = \frac{e^{M^*_{ij}}}{\sum_k e^{M^*_{ik}}} \cdot \frac{e^{M^*_{ij}}}{\sum_l e^{M^*_{lj}}}. \tag{1}$$

Finally, we drop the dustbin row and column from $\hat{M}^*$, and sample $N$ correspondences for subsequent pose estimation. We denote the matching probability of the sampled correspondences as $w_n$, $n \in [1, ..., N]$, which will be used as the weight in the scale-aware Procrustes alignment.

### 3.2 CAMERA POSE ESTIMATION

**Coordinate assignment:** Given the $N$ sampled correspondences, we leverage off-the-shelf monocular depth estimators to assign coordinates to these correspondences, which are then used to estimate the camera pose. Importantly, monocular depth prediction is a geometrically ill-posed problem, as there is no unique mapping from a single image to absolute metric depth. State-of-the-art methods either learn generalizable metric depth from diverse training data (Piccinelli et al., 2025; Zhu et al., 2024), or predict depth that is accurate only up to an unknown per-image scale (Wang & Liu, 2024; Yang et al., 2024a). To generalize across diverse scenarios and image types, our method accommodates both metric and relative depth, and explicitly estimates a scale factor $s$ to convert relative depth into metric space. This scale estimation leverages the metric information, i.e., meters per pixel, available in geo-referenced aerial imagery.

Specically, we denote the metric coordinates of the 2D point associated with the $n$-th aerial feature as $(x_n^A, y_n^A)$, with the origin defined at the center of the aerial image. For the corresponding ground feature, we retrieve its depth from $D$ and use its associated ray direction to compute its 3D location in space. The resulting 3D point is denoted as $(x_n^G, y_n^G, z_n^G)/s$, with the origin defined at the ground camera. Here, $s$ is an unknown scale factor from the ground coordinate system to the aerial metric space, and $s = 1$ when a reliable metric depth model is used. We retain all matched ground points without explicitly selecting them based on their height (z-coordinate). Our ablation study will show that explicitly selecting the topmost point does not improve performance.

**Scale-aware Procrustes alignment:** As shown in Fig. 2 right, once we have the point correspondences $\{(x_n^A, y_n^A), (x_n^G, y_n^G, z_n^G)/s\}$, and their matching probabilities $w_n$, we can compute the rotation $\mathbf{R}$, translation $\mathbf{t}$, and the scale $s$ between the ground points and the metric space aerial points analytically in a differentiable manner using 2D scale-aware Procrustes alignment (Umeyama, 1991)[1].

For simplicity, we denote the planar coordinates[2] of all ground points as $\mathbf{P} = \mathbf{P}^*/s$, assuming they differ from the metric coordinates, $\mathbf{P}^* = \{(x_1^G, y_1^G), ..., (x_n^G, y_n^G)\}$, by an unknown scale $s$. The coordinates of all aerial points are denoted as $\mathbf{Q}$. The objective is to estimate the scale $s$, rotation matrix $\mathbf{R}$ and metric translation $\mathbf{t}$ that satisfy the transformation $\mathbf{Q} = s(\mathbf{R} \cdot \mathbf{P}) + \mathbf{t}$.

First, we compute the weighted centroids of the aerial and ground point sets,

$$\bar{\mathbf{Q}} = \frac{1}{W} \sum_{n=1}^{N} w_n \mathbf{Q}_n, \quad \bar{\mathbf{P}} = \frac{1}{W} \sum_{n=1}^{N} w_n \mathbf{P}_n, \quad \text{with } W = \sum_{n=1}^{N} w_n. \tag{2}$$

In Eq. 2, the better matched points will contribute more to the centroids of the point sets. Then, we center the point sets and compute their covariance matrix $\mathbf{C}$,

$$\mathbf{C} = \sum_{n=1}^{N} w_n \left( \tilde{\mathbf{P}}_n \right) \left( \tilde{\mathbf{Q}}_n \right)^{\top}, \tag{3}$$

where $\tilde{\mathbf{P}}_n$ and $\tilde{\mathbf{Q}}_n$ are the $n$-th points in the centered point sets $\tilde{\mathbf{Q}} = \mathbf{Q} - \bar{\mathbf{Q}}$ and $\tilde{\mathbf{P}} = \mathbf{P} - \bar{\mathbf{P}}$. The covariance matrix $\mathbf{C}$ captures how the centered ground points relate to the centered aerial points, encoding the direction and strength of their mutual variation.

Next, we perform Singular Value Decomposition (SVD) on $\mathbf{C}$, where $\mathbf{C} = \mathbf{U}\mathbf{\Sigma}\mathbf{V}^{\top}$, and the optimal rotation can be recovered as $\mathbf{R} = \mathbf{V}\mathbf{U}^{\top}$, which derives the estimated yaw orientation $o$. The scale and translation from the ground points $\mathbf{P}$ to the aerial points $\mathbf{Q}$ are then computed as,

$$s^* = \frac{\text{Tr}(\mathbf{\Sigma})}{\sum_{n=1}^{N} w_n \left\| \tilde{\mathbf{P}}_n \right\|^2}, \qquad \mathbf{t} = \bar{\mathbf{Q}} - s^*(\mathbf{R} \cdot \bar{\mathbf{P}}). \tag{4}$$

Recall that $\mathbf{P} = \mathbf{P}^*/s$. Substituting this into Eq. 4, the estimated scale $s^*$ can be expressed using the (unscaled) metric coordinates $\mathbf{P}^*$, with $\mathbf{\Sigma}^*$ computed from the SVD between $\mathbf{Q}$ and $\mathbf{P}^*$:

---

[1] In Xia & Alahi (2025); Barroso-Laguna et al. (2024), all coordinates are in metric space, so they use orthogonal Procrustes (Gower, 1975) and ignore scale.

[2] We assume that the ground camera's viewing direction is orthogonal to the gravity direction, since this can be reliably calibrated (Veicht et al., 2024) or obtained from an IMU or accelerometer.

$$s^* = \frac{\text{Tr}(\mathbf{\Sigma}^*/s)}{\sum_{n=1}^{N} w_n \left\| \tilde{\mathbf{P}}^{*n}/s \right\|^2} = \frac{s\,\text{Tr}(\mathbf{\Sigma}^*)}{\sum_{n=1}^{N} w_n \left\| \tilde{\mathbf{P}}^{*n} \right\|^2}. \tag{5}$$

Since $\mathbf{P}^*$ lies in the same metric space as the aerial points $\mathbf{Q}$, and there is no scale difference between them, we have $\text{Tr}(\mathbf{\Sigma}) = \sum_{n=1}^{N} w_n \left\| \tilde{\mathbf{P}}_n \right\|^2$. This leads to $s^* = s$, indicating that the ground points $\mathbf{P}$ must be scaled by $s$ to align them with the metric aerial points $\mathbf{Q}$.

**Practical implication:** This derivation shows that, regardless of the unknown scale of the ground points, scale-aware Procrustes alignment yields a consistent estimate of the camera pose and can recover the scale $s$. Its differentiability makes it particularly well-suited for learning local feature matching through pose supervision, while also enabling inference using only relative depth.

### 3.3 MODEL SUPERVISION

We follow Xia & Alahi (2025) to provide supervision on both the pose and correspondences. For camera pose, we adopt the Virtual Correspondence Error (VCE) loss. Specifically, we define a set of $N_v$ virtual points $\mathbf{P}_v$ in a 2D metric space, where the hyperparameter $l$ controls the spatial extent of the space. We apply both the ground-truth transformation, $\mathbf{R}_{\text{gt}}, \mathbf{t}_{\text{gt}}$, and the estimated transformation, $\mathbf{R}, \mathbf{t}$, to these virtual points. The loss $\mathcal{L}_{\text{VCE}}$ then minimizes the mean Euclidean distance between the corresponding transformed points,

$$\mathcal{L}_{\text{VCE}} = \frac{1}{N_v} \sum \left\| (\mathbf{R}_{\text{gt}} \cdot \mathbf{P}_v + \mathbf{t}_{\text{gt}}) - (\mathbf{R} \cdot \mathbf{P}_v + \mathbf{t}) \right\|_2. \tag{6}$$

When metric depth is available during training, we use the ground-truth pose to find, for the ground points, their corresponding aerial points and compute an infoNCE loss (Oord et al., 2018), $\mathcal{L}_{\text{G2S}}$, to encourage these correspondences. We also find, for the aerial points, their corresponding ground points. However, multiple ground points may exist at different heights but share similar BEV locations. We select the ground point closest to the projected location and compute the $\mathcal{L}_{\text{S2G}}$, using only ground points outside a defined local neighborhood as negatives. The detailed formulation is provided in the Appendix. The total loss is then a weighted combination of the VCE loss and infoNCE losses, $\mathcal{L} = \mathcal{L}_{\text{VCE}} + \beta(\mathcal{L}_{\text{G2S}} + \mathcal{L}_{\text{S2G}})/2$.

## 4 EXPERIMENT

We first introduce the datasets and metrics, followed by implementation details. We then compare our method with prior state-of-the-art vision-based methods and showcase inference using different monocular depth predictors. Next, we present qualitative results on local feature matching and interpretable layout alignment, along with cross-dataset generalization and our ablation studies.

### 4.1 DATASETS AND EVALUATION METRICS

**KITTI** (Geiger et al., 2013) provides forward-facing ground-level images with a limited field of view. Aerial images for KITTI were collected by Shi & Li (2022), who also split the dataset into Training, Test1, and Test2 subsets. Test1 includes images from the same region as the Training set, while Test2 contains images from a different unseen region. We refer to Test1 and Test2 as the same-area and cross-area test sets, respectively. We use the metric depth predictions from DepthAnythingV2 (Yang et al., 2024b) for our experiments on KITTI.

**VIGOR** (Zhu et al., 2021) is a widely used cross-view localization dataset, containing ground-level panoramas and aerial images from four U.S. cities. It also provides same-area and cross-area splits. In the same-area split, training and test images are collected from all four cities, while in the cross-area split, the training and test sets come from two different cities. During training, we adopt a recent metric depth model, Unik3D (Piccinelli et al., 2025), which has shown strong performance on outdoor panoramic images. At test time, we report results using both Unik3D and relative depth

models (Wang & Liu, 2024; Jiang et al., 2021; Wang et al., 2022). Additionally, we include results in the Appendix where both training and testing are performed using only relative depth.

**Metrics:** We report the mean and median localization and orientation errors. Following Xia et al. (2023), we train and test with both known and unknown orientations on VIGOR, and apply orientation noise sampled from $\pm 10°$ or $\pm 180°$ on KITTI. For KITTI, we also decompose localization errors into longitudinal and lateral components based on the driving direction and report the percentage of samples within the defined error thresholds.

## 4.2 Implementation Details

On both datasets, we use AdamW (Loshchilov & Hutter, 2019) with a learning rate of $1 \times 10^{-4}$. Training is conducted on a single H100 GPU with a batch size of 80 for VIGOR and 224 for KITTI. We follow the setting from Xia & Alahi (2025): the temperature parameter $\tau$ in matching score computation is set to 0.1, the aerial feature map $\mathbf{F}_A$ is resized to generate $41 \times 41$ aerial points, $\mathcal{L}_{\text{VCE}}$ uses $N_v = 10 \times 10$ points, with $l$ set to $5\,\text{m}$, and $N = 1024$ correspondences are sampled for pose estimation. For $\mathcal{L}_{\text{S2G}}$, negatives are $1\,\text{m}$ away in planar distance from the projected location. We use $\beta = 1$ for VIGOR and $\beta = 0.1$ for KITTI. RANSAC (Fischler & Bolles, 1981) is applied on VIGOR, but omitted on KITTI as it did not yield improvements in localization accuracy. When using metric depth, we apply a maximum depth threshold of $35\,\text{m}$ for VIGOR and $40\,\text{m}$ for KITTI, setting the matching score of points exceeding this threshold to zero. For relative depth predictors, we apply a fixed initial scale (identified by visually inspecting a few examples) to all relative depth maps and then use the threshold to filter out matches corresponding to sky and distant objects.

## 4.3 Quantitative Results

Table 1: KITTI test results. **Best in bold.** The 'ori.' column shows orientation noise used in training and testing, uniformly sampled within $\pm 10°$ or $\pm 180°$.

| | Ori. | Methods | ↓ Loc. (m) Mean | Median | ↑ Lateral (%) R@1m | R@5m | ↑ Long. (%) R@1m | R@5m | ↓ Orien. (°) Mean | Median | ↑ Orien. (%) R@1° | R@5° |
|---|---|---|---|---|---|---|---|---|---|---|---|---|
| Cross-area | ±10° | GGCVT | - | - | 57.72 | 91.16 | 14.15 | 45.00 | - | - | **98.98** | **100.00** |
| | | CCVPE | 9.16 | 3.33 | 44.06 | 90.23 | 23.08 | 64.31 | **1.55** | **0.84** | 57.72 | 96.19 |
| | | HC-Net | 8.47 | 4.57 | **75.00** | **97.76** | **58.93** | **76.46** | 3.22 | 1.63 | 33.58 | 83.78 |
| | | DenseFlow | 7.97 | 3.52 | 54.19 | 91.74 | 23.10 | 61.75 | 2.17 | 1.21 | 43.44 | 89.31 |
| | | FG² | 7.31 | 4.15 | 37.89 | 85.65 | 21.98 | 60.77 | 3.62 | 2.37 | 23.03 | 77.84 |
| | | Ours | **5.60** | **3.01** | 45.29 | 92.43 | 27.01 | 68.26 | 3.32 | 2.12 | 26.03 | 80.68 |
| | ±180° | SliceMatch | 14.85 | 11.85 | **24.00** | **72.89** | 7.17 | 33.12 | 23.64 | 7.96 | **31.69** | **31.69** |
| | | CCVPE | 13.94 | 10.98 | 23.42 | 60.46 | 11.81 | 42.12 | 77.84 | 63.84 | 3.14 | 14.56 |
| | | Ours | **11.71** | **9.11** | 13.64 | 50.88 | **14.04** | **50.73** | 55.18 | 33.23 | 2.53 | 12.78 |
| Same-area | ±10° | GGCVT | - | - | 76.44 | 98.89 | 23.54 | 62.18 | - | - | **99.10** | **100.00** |
| | | CCVPE | 1.22 | 0.62 | 97.35 | 99.71 | 77.13 | 97.16 | 0.67 | 0.54 | 77.39 | 99.95 |
| | | HC-Net | 0.80 | 0.50 | **99.01** | 99.73 | 92.20 | **99.25** | **0.45** | 0.33 | 91.35 | 99.84 |
| | | DenseFlow | 1.48 | **0.47** | 95.47 | 99.79 | 87.89 | 94.78 | 0.49 | **0.30** | 89.40 | 99.31 |
| | | FG² | **0.75** | 0.51 | 95.81 | 99.66 | **92.50** | 99.05 | 0.93 | 0.66 | 67.27 | 98.91 |
| | | Ours | 1.13 | 0.77 | 93.59 | **99.97** | 71.51 | 98.17 | 1.97 | 1.43 | 36.68 | 92.84 |
| | ±180° | SliceMatch | 7.96 | 4.39 | 49.09 | **98.52** | 15.19 | 57.35 | **4.12** | **3.65** | **13.41** | **64.17** |
| | | CCVPE | 6.88 | 3.47 | 53.30 | 85.13 | 25.84 | 68.49 | 15.01 | 6.12 | 8.96 | 42.75 |
| | | Ours | **1.85** | **1.31** | **62.18** | 97.80 | **59.37** | **97.48** | 9.70 | 6.17 | 9.46 | 41.64 |

**KITTI:** We compare our method with both global descriptor-based methods, CCVPE (Xia et al., 2023) and SliceMatch (Lentsch et al., 2023), and geometry transformation-based methods, GGCVT (Shi et al., 2023), HC-Net (Wang et al., 2024b), DenseFlow (Song et al., 2024), and FG² (Xia & Alahi, 2025). As shown in Tab. 1, our method sets a new state-of-the-art in mean and median localization errors on the cross-area test under both $\pm 10°$ and $\pm 180°$ orientation noise. On the same-area test, although our method has higher error under $\pm 10°$ noise, it shows a large improvement over the previous state-of-the-art method in the more challenging $\pm 180°$ setting, reducing the mean error from $6.88\,\text{m}$ to $1.85\,\text{m}$. For orientation prediction, our method is less accurate than the state of the art. Since our method computes orientation analytically from local feature correspondences, it cannot take full advantage of the prior that most ground images in KITTI are

aligned with the road direction. Under $\pm 10°$ noise, our performance is comparable to FG$^2$, while under $\pm 180°$ noise, our method outperforms CCVPE but falls short of SliceMatch.

**VIGOR:** Overall, our method demonstrates strong and consistent performance in both localization and orientation estimation on the same-area and cross-area test sets (see Tab. 2). For instance, while FG$^2$ attains lower localization error when the orientation is known, our method significantly outperforms it in the more challenging unknown orientation setting. Compared to the previous state of the art under unknown orientation (Xia et al., 2023), our method achieves comparable localization accuracy, showing slightly higher mean error on the same-area test set but lower mean error on the cross-area test set. Notably, for panoramic images, the richer matchable information between ground and aerial views leads to substantial gains, with our method achieving the lowest mean orientation error in both same-area and cross-area tests.

Table 2: VIGOR test results. **Best in bold.** Training uses metric depth from Unik3D (Piccinelli et al., 2025). The row 'ours' reports results with metric depth from the same model, while 'ours-xxx' shows results with different relative depth inputs. Relative depth predictors, BiFuse++ (Wang et al., 2022) and UniFuse (Jiang et al., 2021), are provided by (Wang & Liu, 2024). 'Ours-Unik3D$_{rel}$' denotes the study where metric depth is manually scaled by an arbitrary factor.

| Ori. | Methods | Cross-area | | | | Same-area | | | |
| | | ↓ Localization (m) | | ↓ Orientation (°) | | ↓ Localization (m) | | ↓ Orientation (°) | |
| | | Mean | Median | Mean | Median | Mean | Median | Mean | Median |
|---|---|---|---|---|---|---|---|---|---|
| Unknown | SliceMatch | 7.22 | 3.31 | 25.97 | 4.51 | 6.49 | 3.13 | 25.46 | 4.71 |
| | CCVPE | 5.41 | **1.89** | 27.78 | 13.58 | **3.74** | **1.42** | 12.83 | 6.62 |
| | DenseFlow | 7.67 | 3.67 | 17.63 | 2.94 | 4.97 | 1.90 | 11.20 | **1.59** |
| | FG$^2$ | 10.02 | 8.14 | 31.41 | 5.45 | 8.95 | 7.32 | 15.02 | 2.94 |
| | Ours | **4.23** | 2.09 | **11.67** | **2.21** | 3.94 | 1.78 | **9.54** | 2.00 |
| | Ours-BiFuse++ | 4.43 | 2.26 | 12.27 | 2.47 | 4.12 | 1.94 | 10.13 | 2.31 |
| | Ours-UniFuse | 4.36 | 2.21 | 12.10 | 2.43 | 4.03 | 1.89 | 9.81 | 2.21 |
| Known | SliceMatch | 5.53 | 2.55 | - | - | 5.18 | 2.58 | - | - |
| | CCVPE | 4.97 | 1.68 | - | - | 3.60 | 1.36 | - | - |
| | GGCVT | 5.16 | 1.40 | - | - | 4.12 | 1.34 | - | - |
| | DenseFlow | 5.01 | 2.42 | - | - | 3.03 | **0.97** | - | - |
| | HC-Net | 3.35 | 1.59 | - | - | 2.65 | 1.17 | - | - |
| | FG$^2$ | **2.41** | **1.37** | - | - | **1.95** | 1.08 | - | - |
| | Ours | 3.43 | 1.90 | - | - | 3.06 | 1.59 | - | - |
| | Ours-Unik3D$_{rel}$ | 3.43 | 1.90 | - | - | 3.06 | 1.59 | - | - |
| | Ours-BiFuse++ | 3.56 | 2.02 | - | - | 3.17 | 1.69 | - | - |
| | Ours-UniFuse | 3.54 | 2.01 | - | - | 3.14 | 1.67 | - | - |

**Inference with relative depth:** We evaluate the consistency of our pose estimation against different depth predictors in inference through two additional experiments on VIGOR. The first one applies arbitrary scale factors to the metric depth predicted by Unik3D. We vary the scale from 0.001 to 1000 and observe that the resulting variation in both mean and median localization error remains below 1 cm ('Ours-Unik3D$_{rel}$' in Tab. 2), demonstrating strong scale invariance. This robustness is due to both the accurate correspondence matching and the spatial distribution of the matched points.

Second, we evaluate a practical scenario where only a relative depth model (Wang et al., 2022; Jiang et al., 2021; Wang & Liu, 2024) is available at inference. We plug in different relative depth models with our estimated correspondences, without retraining or finetuning. As shown in the bottom two rows of the known and unknown orientation entries in Tab. 2, relative depth increases localization error by less than 0.2 m. This flexibility in depth predictors highlights the practicality and robustness of our method for real-world deployment, especially when a compact depth predictor is required.

## 4.4 INTERPRETABILITY

**Local feature matching:** Since our pose is computed analytically from the estimated correspondences, these correspondences directly reflect localization quality, an interpretability advantage absent in most prior cross-view localization methods. Although FG$^2$ also estimates correspondences, its localization fails when the orientation is unknown, whereas our method remains accurate.

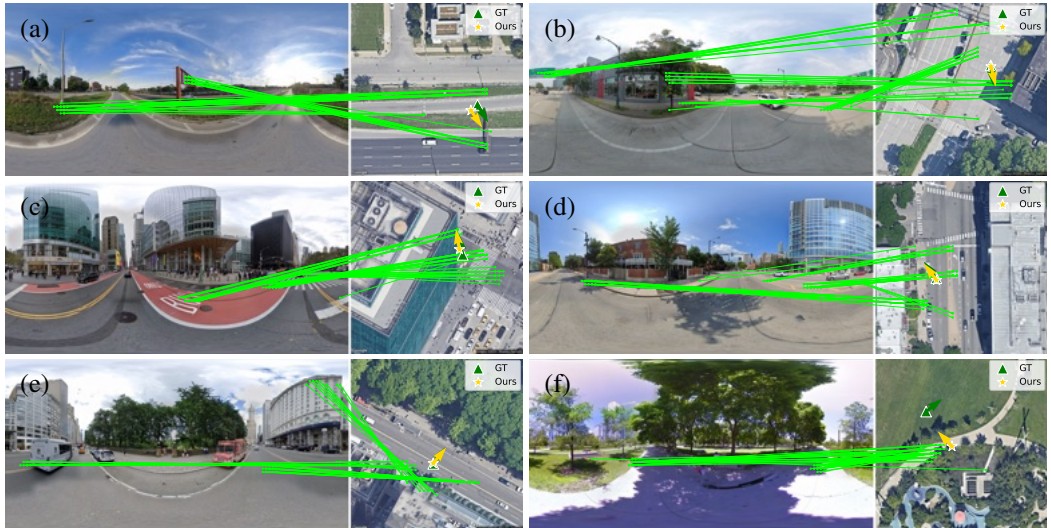

Figure 3: Local feature matching results on the VIGOR same-area test set under unknown orientation. We visualize the top 50 correspondences, ranked by matching score.

As shown in Fig. 3 (a) and (b), when road markings are repetitive, e.g., on highways, our method leverages other landmarks such as overhead road signs and streetlights for localization, demonstrating that the matching is not solely appearance-based but also reflects strong semantic understanding. In (c), the partially occluded 'bus' marking is difficult even for humans to identify, yet our model matches it correctly. In (d), both streetlights and road markings serve as anchors for localization, while (e) shows that without restricting matches to object tops, our method aligns the upper facades of buildings in the ground view with rooftops in the aerial view. Finally, (f) shows a failure case, though interestingly our method predicts a more reasonable position aligned with the visible path, whereas the ground truth appears erroneous. More VIGOR and KITTI results are in the Appendix.

**Outlier detection:** The ability to estimate correspondences makes our method well-suited for outlier detection using RANSAC. For each test sample, we record the number of inlier correspondences and use it for outlier detection. As shown in Fig. 4, the pose error decreases sharply as the inlier ratio rises from 10% to 50%, followed by a slower decline after 50%. This trend reveals a strong negative correlation between inlier ratio and pose error, consistent with the expectation that more accurate correspondences yield more accurate pose estimates.

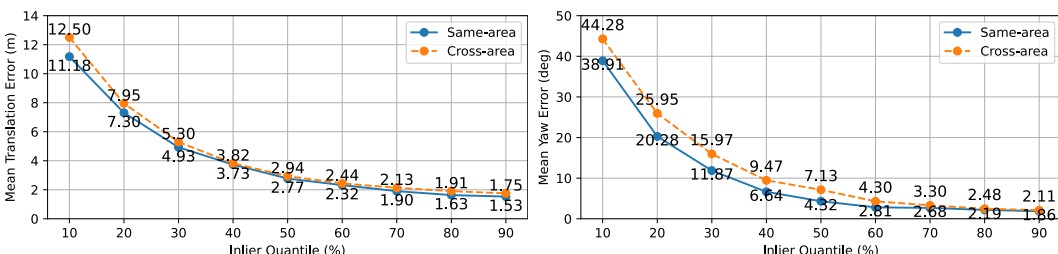

Figure 4: Outlier detection using RANSAC on VIGOR same/cross-area test sets.

**Ground-aerial layout alignment:** With a relative depth predictor (Wang et al., 2022), our method estimates both the camera pose and the scale between relative depth and the aerial metric space. As shown in Fig. 5, the rotated, translated, and scaled ground layout can be overlaid onto the aerial image, providing a clear and interpretable visual cue of localization quality. For example, (a) and (b) show precise alignment of the projected intersection and building with those in the aerial view,

corresponding to accurate localization. Interestingly, (c) also shows good alignment, revealing an error in the ground-truth location: the vehicle should be in front of the zebra crossing rather than on it. (d) shows a localization failure, which is easily identifiable from the misalignment. The strong correlation between alignment and localization quality offers a powerful tool for interpretability.

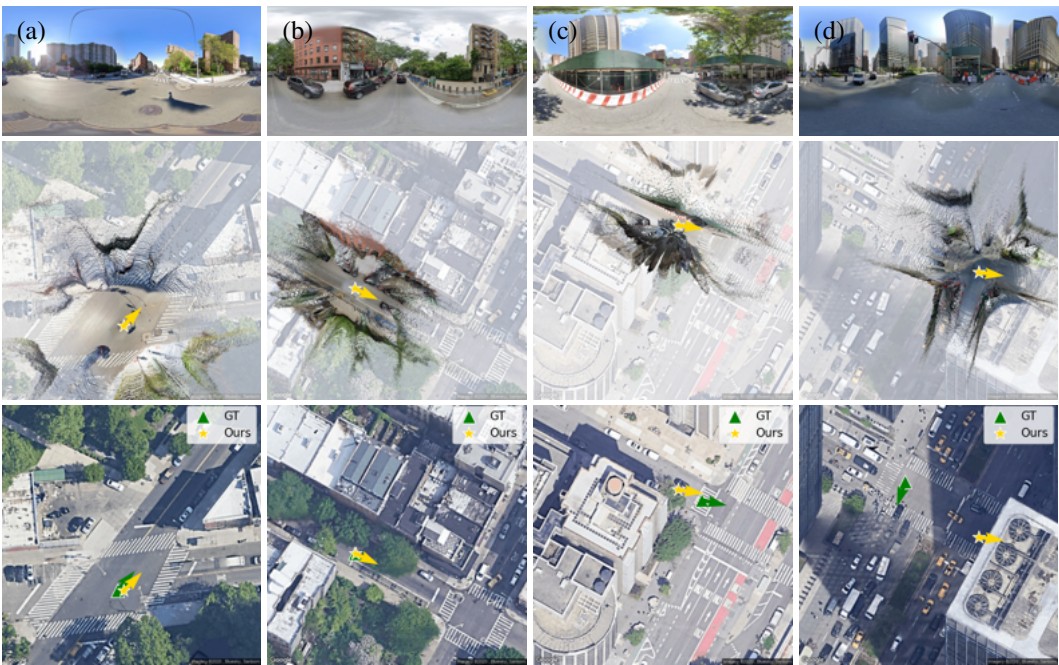

Figure 5: Ground layout overlaid on the aerial image after applying the predicted rotation, translation, and scale transformations. The alignment directly reflects localization quality: the first three examples show successful localization, while the last one illustrates a failure case. Notably, the alignment in example (c) helped us to identify the error in ground truth location.

## 4.5 CROSS-DATASET GENERALIZATION

Additionally, we demonstrate the strong generalization capability of our model by directly applying the model trained on VIGOR to the CVACT dataset (Liu & Li, 2019). CVACT is a cross-view image retrieval dataset collected in Canberra, Australia, where the rural and natural landscapes present a significant domain gap compared to VIGOR. Notably, CVACT lacks precise localization labels (Shi et al., 2022). Therefore, we provide local feature matching results and evaluate localization performance by overlaying the transformed ground layout onto the aerial image.

In general, our model exhibits strong generalization capability across datasets collected in different countries. As shown in Fig. 6, our model establishes reasonable feature correspondences. In samples (a) and (c), features extracted along trees in the ground view accurately match the corresponding blobs in the aerial view. In samples (b) and (d), the curbs and road markings in the ground view are correctly matched to those in the aerial view. Reliable feature matching leads to accurate localization. As shown, the transformed ground layout consistently aligns well with the aerial image across all samples. In (a) and (b), road markings and lane boundaries projected from the ground view form seamless continuations of those in the aerial view. In (c) and (d), both ground-level structures (e.g., curbs) and above-ground objects (e.g., buildings and trees) exhibit strong alignment.

## 4.6 ABLATION STUDY

Our main ablation study focuses on two key design choices: (1) the coordinate assignment strategy (Sec. 3.2), comparing using all points within the depth range versus only the topmost points along height; and (2) the effect of ignoring scale in Procrustes alignment (reducing our formulation to

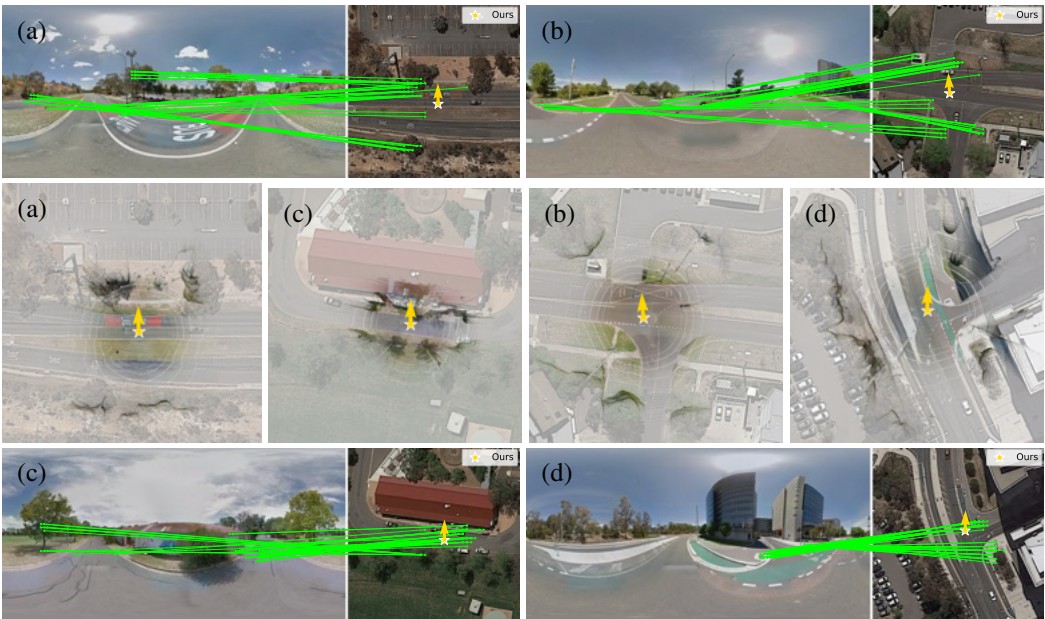

Figure 6: Direct generalization to the CVACT dataset. We visualize the top 50 correspondences, ranked by matching score, and overlay the ground layout on the aerial image after using the predicted rotation, translation, and scale transformations. We use the metric depth prediction from Unik3D.

the orthogonal Procrustes of Xia & Alahi (2025); Barroso-Laguna et al. (2024)). As shown in Tab. 3, explicitly selecting the topmost points does not improve localization accuracy. The side view of an object's top does not necessarily exhibit the most consistent semantics with the aerial image. Therefore, enforcing the use of topmost points may hinder cross-view localization. For scale-awareness, excluding scale forces the model to align features strictly to metric depth predictions, which are imperfect. Allowing the model to also estimate scale from correspondences yields more accurate and robust matching. Moreover, without modeling scale, the method cannot support relative depth predictors, as orthogonal Procrustes assumes that all coordinates lie in the same scale space. Additional ablation study on other hyperparameters in both inference and training is included in Appendix. Sec. A.

Table 3: Ablation study on VIGOR same-area validation set with unknown ori. **Best in bold.**

| Method design choices | Mean loc. (m) | Median loc. (m) | Mean ori. (°) | Median ori. (°) |
|---|---|---|---|---|
| (1) Top points only | 3.95 | 1.78 | 9.37 | **1.77** |
| (2) Not consider scale | 5.47 | 2.75 | 19.92 | 4.45 |
| Ours (all points, scale-aware) | **3.86** | **1.75** | **9.30** | 1.93 |

## 5 CONCLUSION

We propose a simple, accurate, and interpretable fine-grained cross-view localization method that matches local features between ground and aerial images. Our method learns correspondences from camera pose supervision and leverages monocular depth predictors with scale-aware Procrustes alignment to estimate the camera pose and recover the scale of relative depth. Experiments demonstrate state-of-the-art performance in challenging scenarios, including cross-area generalization and unknown camera orientation. Furthermore, our local feature matching enhances interpretability, enabling visual identification of erroneous predictions as well as RANSAC-based outlier detection.

**Reproducibility statement:** Our method combines a deep learning model with a classic algorithmic component, scale-aware Procrustes alignment. The deep learning component is described in Sec. 3.2, while the derivation of the scale-aware Procrustes alignment is detailed in Sec. 3.2 with a complete proof provided in the Appendix, ensuring that the theoretical foundation of our approach is transparent and verifiable. The implementation is lightweight and relies only on standard components, with all hyperparameters and training settings fully described in Sec. 4.2. To further support reproducibility, we will release our code together with configuration files and evaluation scripts, enabling the community to reproduce all experiments and results reported in this paper.

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

APPENDIX

Here we provide supplementary material to support the main paper:

## A ADDITIONAL ABLATION STUDY

We conduct additional ablation studies on the VIGOR same-area validation set with unknown orientation, analyzing the effects of various hyperparameters in both inference and training.

During inference, we vary the number of sampled correspondences $N$, the number of aerial points, the resolution (meters per pixel) of the input aerial image, and the use of RANSAC in our trained model using the default settings.

During training, we investigate the influence of various hyperparameters, including the number of sampled correspondences $N$, the number of aerial points, the feature map resolution, and the temperature parameter $\tau$. Additionally, we compare our formulation (image-plane matching + scale-aware Procrustes alignment) with two homography-based formulation: (1) Replacing image-plane matching with BEV-plane matching by first transforming the ground images into a BEV, and then feed it to the same feature extractor, matcher, and Procrustes alignment. (2) Replacing the scale-aware Procrustes alignment with homography for pose estimation.

### A.1 ABLATION STUDY ON INFERENCE SETTINGS

**Number of sampled correspondences $N$:** Our default setting uses $N = 1024$ (following Xia & Alahi (2025)) during inference. If RANSAC is used, we sample only 3 correspondences per RANSAC loop to compute the pose and scale. In this case, we vary $N$ only in inference: a model trained with $N = 1024$ is evaluated with $N = 256, 512, 1024,$ and $2048$ (without RANSAC).

As shown in Tab. 4, the number of correspondences $N$ does not have a strong impact on performance. When varying $N$ during inference, the difference in mean localization error is less than 0.2 m for $N = 512, 1024, 2048$. The error increases more noticeably when a smaller $N$, such as 256, is used.

**Number of aerial points:** Next, we examine how the number of aerial points affects performance. Our default setting, following Xia & Alahi (2025), uses $41 \times 41$ aerial points for fair comparison. As shown in Tab. 5, our model is robust to small variations in the number of aerial points. When more than $36 \times 36$ points are used, the change in mean localization error is less than 0.2 m. However, when the number of points is significantly reduced (e.g., $31 \times 31$), performance degrades noticeably.

**Aerial image resolution:** We also study the effect of varying aerial image resolution in inference. As shown in Tab. 6, our model is robust to such variations. All aerial images are first processed by the pre-trained DINOv2 feature extractor. Regardless of input resolution, we consistently sample a fixed

Table 4: Ablation on the number of sampled correspondences $N$. **Best in bold.**.

| Mode | N | Localization (m) | | Orientation (°) | |
|---|---|---|---|---|---|
| | | Mean ↓ | Median ↓ | Mean ↓ | Median ↓ |
| Inference | 2048 | 4.66 | 2.88 | **11.13** | **3.92** |
| | 1024 (default) | **4.60** | **2.81** | 11.19 | 4.04 |
| | 512 | 4.71 | 2.91 | 12.06 | 4.20 |
| | 256 | 4.90 | 2.98 | 12.97 | 4.78 |

Table 5: Ablation on the number of aerial points. **Best in bold.**

| **Number of Aerial Points** | **Localization (m)** | | **Orientation (°)** | |
|---|---|---|---|---|
| | **Mean ↓** | **Median ↓** | **Mean ↓** | **Median ↓** |
| 31×31 | 4.27 | 2.16 | 10.01 | 2.49 |
| 36×36 | 3.98 | 1.88 | 9.38 | 2.10 |
| 41×41 (default) | 3.86 | 1.75 | 9.30 | 1.93 |
| 46×46 | 3.86 | 1.72 | 9.31 | 1.91 |
| 51×51 | **3.82** | **1.70** | **9.18** | **1.90** |
| 56×56 | 3.91 | 1.68 | 9.61 | 1.97 |

$41 \times 41$ grid of aerial points from the feature map. Since a downsampled aerial image still covers the same geographic area, the sampled points correspond to the same geo-locations, independent of resolution. Thus, any effect on pose estimation accuracy mainly stems from changes in DINOv2 features due to downsampling in lower-resolution inputs.

Table 6: Ablation on aerial image resolution. **Best in bold.**

| Resolution | Localization (m) | | Orientation (°) | |
|---|---|---|---|---|
| | Mean ↓ | Median ↓ | Mean ↓ | Median ↓ |
| 630×630 (default) | 3.86 | 1.75 | 9.30 | **1.93** |
| 574×574 | **3.77** | **1.71** | 9.52 | 1.95 |
| 518×518 | 3.82 | 1.76 | **9.17** | 1.94 |
| 448×448 | 3.90 | 1.77 | 9.49 | 1.97 |
| 392×392 | 3.99 | 1.84 | 10.22 | 2.13 |
| 336×336 | 4.17 | 1.96 | 10.36 | 2.28 |

**RANSAC:** Finally, we report results of in inference with or without RANSAC . As shown in Tab. 7, using RANSAC yields a ∼0.74 m reduction in mean localization error.

## A.2 Ablation study on training settings

We first investigate the impact of several hyperparameters involved in our local feature matching during training, including the number of sampled correspondences, the number of aerial points, and the temperature parameter. After this, we vary our formulation by using homography-transformed inputs or a homography-based pose estimator.

**Number of sampled correspondences $N$:** Our default setting uses $N = 1024$ (following Xia & Alahi (2025)) during training. In this case, we vary $N$ in both training and inference: validation results (with RANSAC) of models trained with $N = 256, 512, 1024$, and $2048$. As shown in Tab. 8, the model remains robust to different $N$ during training, as long as $N$ is not too small (e.g., $N \leq 256$).

**Number of aerial points & Feature Map Resolution:** We train additional models using $31 \times 31$, $36 \times 36$, $46 \times 46$, $51 \times 51$, $56 \times 56$ aerial points. Meanwhile, to investigate the impact of feature map resolution, we also trained a model that upsamples the aerial DINO feature map from its original

Table 7: Ablation on RANSAC. **Best in bold.**

| RANSAC | Localization (m) | | Orientation (°) | |
|---|---|---|---|---|
| | Mean ↓ | Median ↓ | Mean ↓ | Median ↓ |
| Without RANSAC | 4.60 | 2.81 | 11.19 | 4.04 |
| With RANSAC | **3.86** | **1.75** | **9.30** | **1.93** |

Table 8: Ablation on the number of sampled correspondences $N$. **Best in bold..**

| Mode | N | Localization (m) | | Orientation (°) | |
|---|---|---|---|---|---|
| | | Mean ↓ | Median ↓ | Mean ↓ | Median ↓ |
| Training | 2048 | 3.91 | 1.76 | **9.25** | **1.73** |
| | 1024 (default) | **3.86** | **1.75** | 9.30 | 1.93 |
| | 512 | 4.12 | 1.80 | 9.71 | 1.80 |
| | 256 | 4.31 | 1.90 | 11.27 | 2.17 |

$45 \times 45$ resolution to $180 \times 180$ using two stages of bilinear interpolation followed by convolutional layers. We then sample a $41 \times 41$ grid of points from this higher-resolution feature map.

As shown in Tab. 9, higher resolution of aerial points provides only a modest improvement in performance (0.16 meters reduction in mean localization error). However, such setting substantially increases memory usage and training time. Notably, Tab. 10 indicates that this marginal improvement mainly comes from RANSAC, which benefits from a larger pool of correspondences. When inferring without RANSAC, the $41 \times 41$ setting even performs slightly better. By contrast, upsampling the feature maps does not lead to any measurable performance improvement.

Table 9: Ablation on different resolution configurations. **Best in bold.**

| Number of aerial points ($N$) | Mean loc. (m) | Median loc. (m) | Mean ori. (°) | Median ori. (°) |
|---|---|---|---|---|
| $31 \times 31$ | 4.22 | 1.90 | 9.68 | 1.99 |
| $36 \times 36$ | 4.01 | 1.82 | 9.42 | 1.91 |
| $41 \times 41$ (default) | 3.86 | 1.75 | 9.30 | 1.93 |
| $41 \times 41$ (upsampled feature) | 4.55 | 1.94 | 11.46 | 1.96 |
| $46 \times 46$ | 3.90 | 1.72 | 9.07 | 1.72 |
| $51 \times 51$ | 3.75 | 1.67 | 8.76 | 1.73 |
| $56 \times 56$ | **3.70** | **1.65** | **8.75** | **1.69** |

**Temperature Parameter $\tau$:** This parameter directly affects the matching score matrix $M^*$. Following Barroso-Laguna et al. (2024); Sun et al. (2021); Xia & Alahi (2025), we adopt the commonly used setting $\tau = 0.1$. To further investigate its impact, we perform a grid search over $\tau \in 0.5, 0.1, 0.05, 0.01$. As shown in Tab. 11, $\tau = 0.1$ and $\tau = 0.05$ yield very similar performance, while both larger and smaller values lead to a noticeable degradation in accuracy.

**BEV-plane matching:** Transforming ground images into BEV via a homography typically introduces ray-directional distortions, which degrade correspondence quality when matching these BEV-transformed views directly to the aerial image. To validate this, we construct a bird's-eye-view (BEV) variant of Loc$^2$ by first transforming the ground-level images into BEV representations and then feeding the transformed images into the same feature extractor and matching head, followed by scale-aware Procrustes alignment using the same loss functions. We adopted the BEV transformation from HC-Net (Wang et al., 2024b) with its default hyperparameters.

As shown in the Tab. 12, the BEV variant yields substantially worse performance. The BEV transformation inevitably introduces distortions to elevated or above-ground structures. More importantly, such structures often occupy a large portion of the input image, restricting matching to a small undistorted region or forcing the model to match features on distorted objects, which ultimately degrades performance (see Fig. 7).

Table 10: Different resolution configurations (inference without RANSAC). **Best in bold.**

| Number of aerial points ($N$) | Mean loc. (m) | Median loc. (m) | Mean ori. (°) | Median ori. (°) |
| --- | --- | --- | --- | --- |
| $41 \times 41$ (default) | **4.60** | **2.81** | **11.19** | **4.04** |
| $51 \times 51$ | 4.70 | 2.89 | 11.38 | 4.03 |
| $56 \times 56$ | 4.68 | 2.91 | 11.59 | 3.92 |

Table 11: Different values for the temperature parameter. **Best in bold.**

| Temperature $\tau$ | Mean loc. (m) | Median loc. (m) | Mean ori. (°) | Median ori. (°) |
| --- | --- | --- | --- | --- |
| 0.5 | 9.58 | 6.18 | 39.03 | 15.33 |
| 0.1 (default) | 3.86 | **1.75** | 9.30 | **1.93** |
| 0.05 | **3.78** | 1.80 | **9.25** | 2.17 |
| 0.01 | 4.58 | 2.46 | 13.70 | 4.78 |

**Homography-based pose estimation:** We next replace our pose estimation module with a homography-based formulation instead of a 2D similarity transformation. Specifically, we use the sampled correspondences to estimate a homography matrix and supervise it through translation and orientation, following a strategy similar to HC-Net (Wang et al., 2024b).

As shown in the Tab. 12, the homography-based formulation leads to a significant drop in localization accuracy. A homography has eight degrees of freedom and, beyond translation and rotation, can model anisotropic scaling and shear. As a result, it can disregard the contour and shape information encoded in the (relative) depth prior. Meanwhile, it also reduces interpretability: the inferred ground layout may become skewed or non-uniformly scaled across different directions in order to satisfy the point correspondences in the aerial view.

## B    TEST ON NON-ROAD FACING SAMPLES ON KITTI

In Sec. 4.3, we noted that our method computes orientation from correspondences, making it difficult to exploit the prior that most ground images in KITTI are aligned with the road direction. Here, we provide an additional quantitative comparison between our method and CCVPE on manually selected KITTI samples that are not aligned with the road direction, denoted as *non-road*.

As shown in Tab. 13, our method consistently outperforms CCVPE (which directly regresses orientation and can easily capture the road-facing bias). This setting better reflects real-world scenarios, where the orientation can be arbitrary and no pre-processing (such as rotating the aerial image for rough alignment) is required. When comparing the "non-road" subset to the full test set (Tab. 1), both CCVPE and our method exhibit increased orientation errors. However, the increase is significantly smaller for our method, indicating greater robustness to challenging viewpoints. Interestingly, we observe that localization errors in the non-road subset are lower than those on the full set. This is likely due to reduced location ambiguity at intersections compared to the more monotonous structure of road-aligned scenes.

## C    ADDITIONAL LOCAL FEATURE MATCHING RESULTS

First, we present additional results on local feature matching. Fig. 8 shows success cases where the pose estimation is accurate, while Fig. 10 presents failure cases.

**Success cases:** Fig. 8 presents results on VIGOR. In (a)–(f), we demonstrate that our model effectively matches various types of road markings across views, including painted text (a)–(d) and lines (e)–(f). In general, road markings are distinctive across views, and accurately matching them significantly aids localization. Besides road markings, our method also leverages other landmarks, such as overhead road signs and streetlights, to help anchor the keypoint locations.(Fig. 8 (g)–(h)). In urban areas, there is typically rich visual information available for matching across ground and aerial views (Fig. 8 (i)–(l)), including buildings, zebra crossings, and lane lines. Our model effectively exploits all of these cues for localization.

Table 12: Comparison between the proposed Loc$^2$ and its BEV variant. **Best in bold.**

| Method design choices | Mean loc. (m) | Median loc. (m) | Mean ori. (°) | Median ori. (°) |
|---|---|---|---|---|
| BEV-plane matching | 9.49 | 8.20 | 53.42 | 31.84 |
| Homography pose estimation | 14.46 | 13.43 | 90.98 | 91.17 |
| Ours | **3.86** | **1.75** | **9.30** | **1.93** |

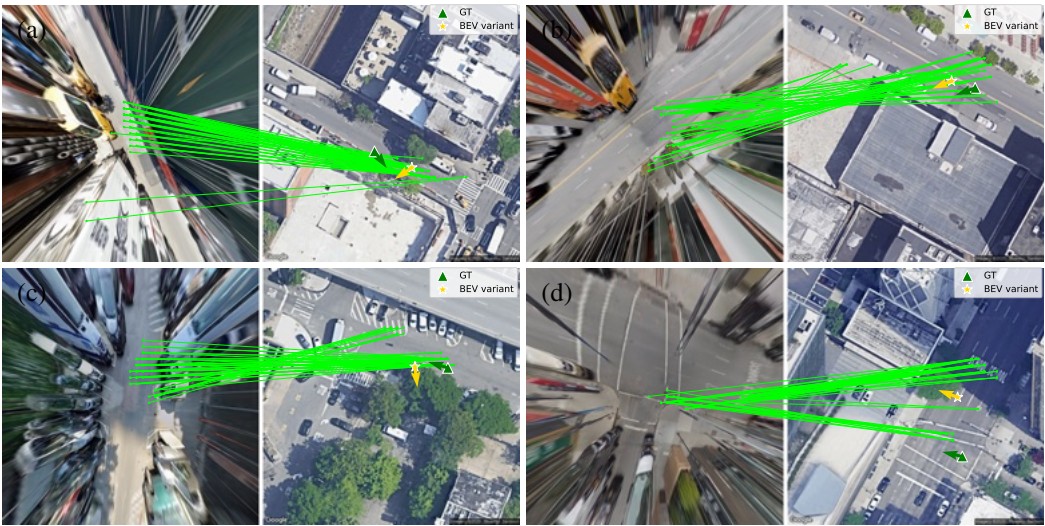

Figure 7: Matching results of the BEV variant of our method on the VIGOR same-area test set under unknown orientation. We visualize the top 50 correspondences, ranked by matching score.

Fig. 9 presents results on KITTI. Our method primarily leverages structures such as roads, sidewalks, and fences in front of the vehicle to establish correspondences with the aerial view.

**Failure cases:** Typical failure cases are shown in Fig. 10. When there is dense vegetation (Fig. 10 (a),(b)), the matchable information across views is very limited, making precise localization extremely challenging, even for humans. Interestingly, in Fig. 10 (b), our method still correctly matches the gap between two buildings in the ground view to the corresponding gap in the aerial view. In Fig. 10 (c), the two narrow roads with trees and a zebra crossing in front appear visually similar. At the true location, a tree partially blocks the zebra crossing in the aerial view, making it harder to distinguish. As a result, our method matches the zebra crossing to an incorrect one, leading to an erroneous location estimate. In Fig. 10 (d), it is nearly impossible to determine the exact location over the water due to the lack of distinctive features. Finally, Fig. 10 (e)–(h) shows that occlusions from dense high-rise buildings make cross-view matching more difficult. Importantly, we believe these scenarios represent common challenges for most cross-view localization methods, rather than limitations specific to our approach. Furthermore, our local feature matching framework provides strong interpretability, enabling straightforward identification of such failure cases and supporting RANSAC-based outlier filtering.

## D ADDITIONAL RESULTS ON CROSS-DATASET GENERALIZATION.

In addition to Sec. 4.5, we present more qualitative results on cross-dataset generalization. As shown in Fig. 11, our model can establish reliable correspondences on buildings (a), road markings (b, c, g, f), and trees (d, e). The layout alignments also demonstrate our superior localization quality with recovered rotation, translation and scale.

Table 13: Test on non-road facing samples on KITTI. **Best in bold.**

| Orientation | Setting | | Localization (m) | | Orientation (°) | |
|---|---|---|---|---|---|---|
| | Test Set | Method | Mean ↓ | Median ↓ | Mean ↓ | Median ↓ |
| Unknown | Cross-area (non-road) | CCVPE | 8.33 | 5.99 | 89.78 | 93.97 |
| | Cross-area (non-road) | Ours | **7.04** | **5.00** | **58.97** | **47.98** |
| | Same-area (non-road) | CCVPE | 3.19 | 1.81 | 26.10 | 16.88 |
| | Same-area (non-road) | Ours | **1.17** | **1.01** | **9.79** | **8.00** |

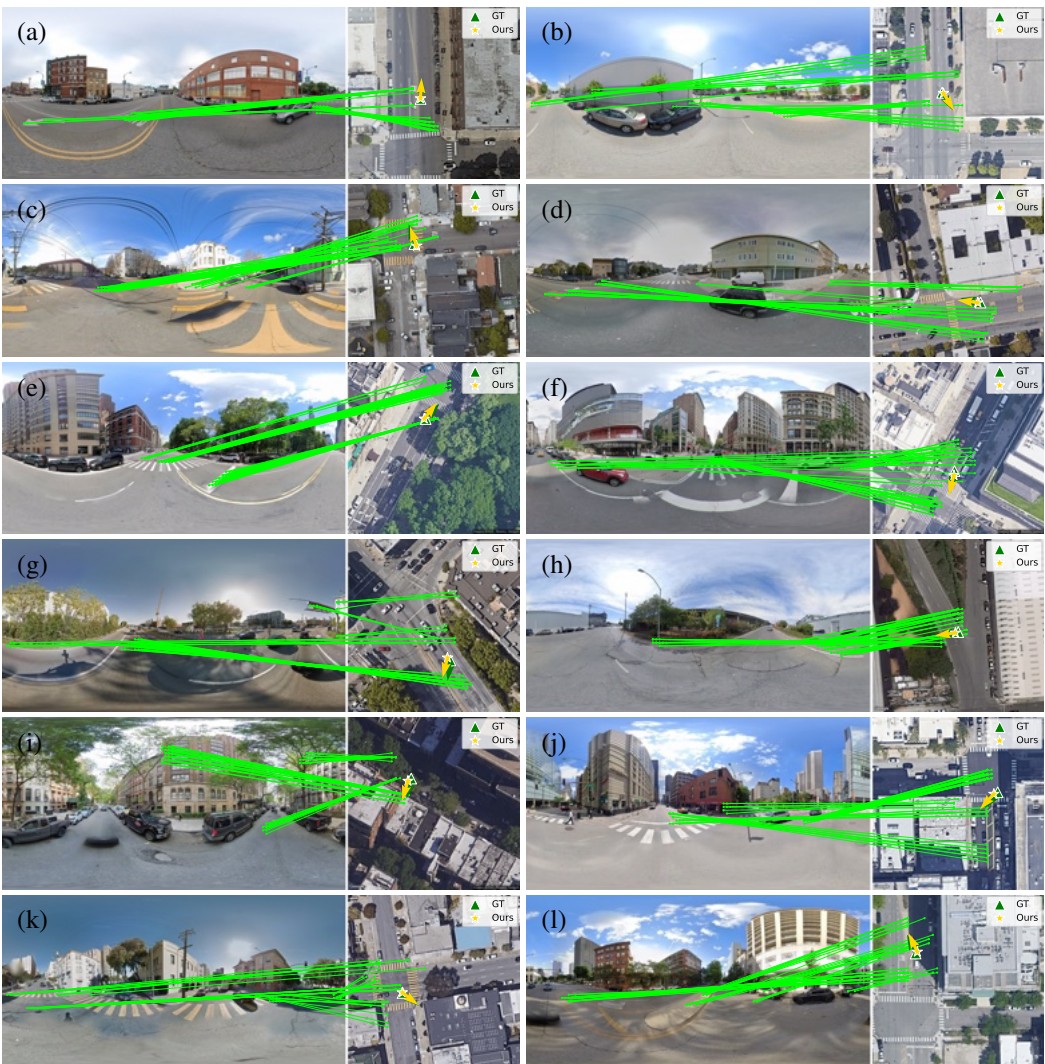

Figure 8: Success cases for localization on the VIGOR same-area test set under unknown orientation. We visualize the top 50 correspondences, ranked by matching score.

# E ADDITIONAL RESULTS ON USING RELATIVE DEPTH IN INFERENCE

We conduct additional experiments on using relative depth at inference time for our model (trained with metric depth predictions from UniK3D (Piccinelli et al., 2025)). Specifically, we use Bi-Fuse++ (Wang et al., 2022) as our relative depth predictor for these experiments.

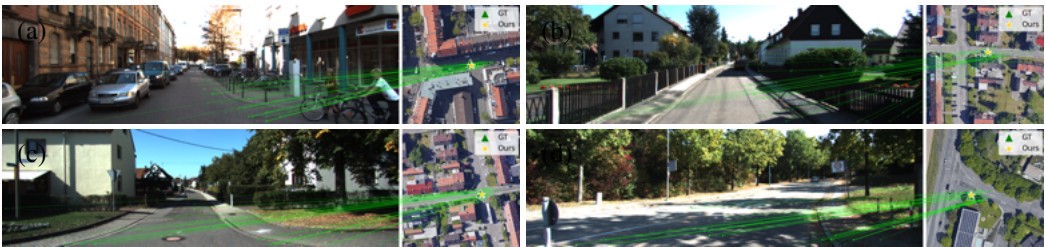

Figure 9: Success cases for localization on the KITTI same-area test set under $\pm 10°$ orientation noise. We visualize the top 50 correspondences, ranked by matching score

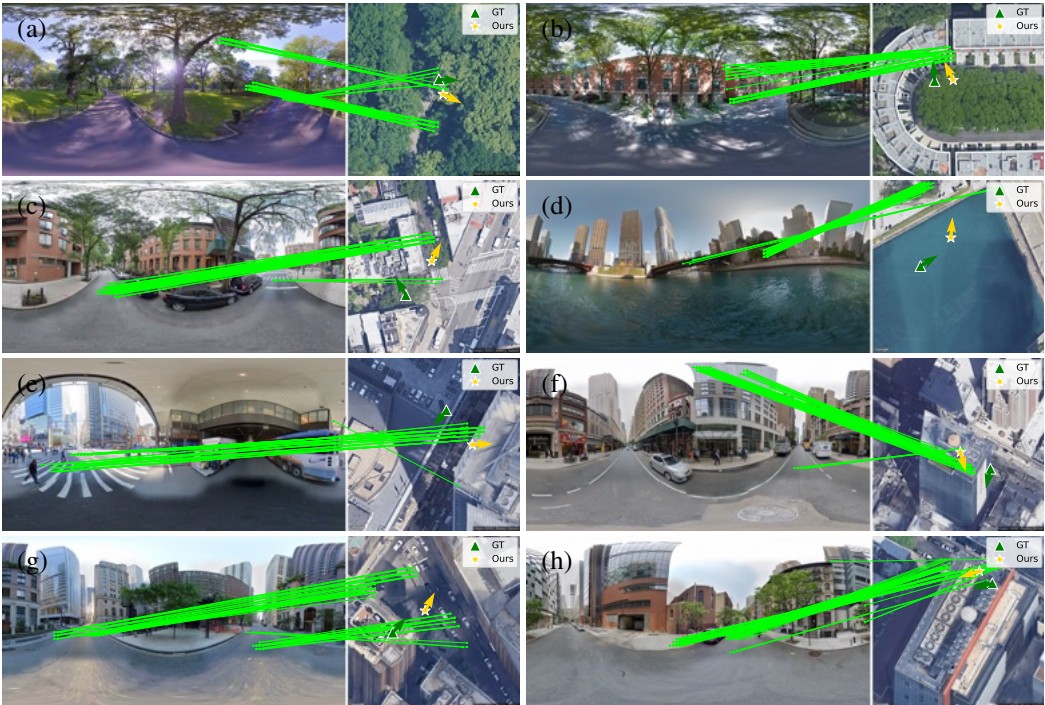

Figure 10: Failure cases for localization on the VIGOR same-area test set under unknown orientation. We visualize the top 50 correspondences, ranked by matching score.

**Initial scale and maximum depth threshold:** As mentioned in Sec. 4.2, when using relative depth, we apply an initial scaling factor to all relative depth maps to bring the values into a reasonable depth range, determined by visually inspecting a few examples. We then apply a predefined maximum depth threshold, the same one used for the metric depth model, to filter out matches corresponding to sky and distant objects. In practice, the initial scale and maximum depth threshold can be determined for each dataset by visually inspecting a few examples. Here, we study how consistent the model's predictions are when these settings are varied.

First, we evaluate the model's robustness against different scales of the relative depth. We apply an additional scaling factor on top of the initial scale. To ensure the same objects are kept for pose estimation, the same scaling factor is also applied to the maximum depth threshold. As shown in Fig. 12, the test errors remain stable across different scaling factors during inference. This observation is consistent with our findings on applying scaling factors to metric depth in Sec. 4.3. Therefore, in practice, when using relative depth, people do not need to carefully select an initial scale.

We then evaluate the effect of different maximum depth thresholds. Our default setting uses a threshold of 35 m, and we additionally test thresholds of 25 m and 45 m. Note that although a 35 m

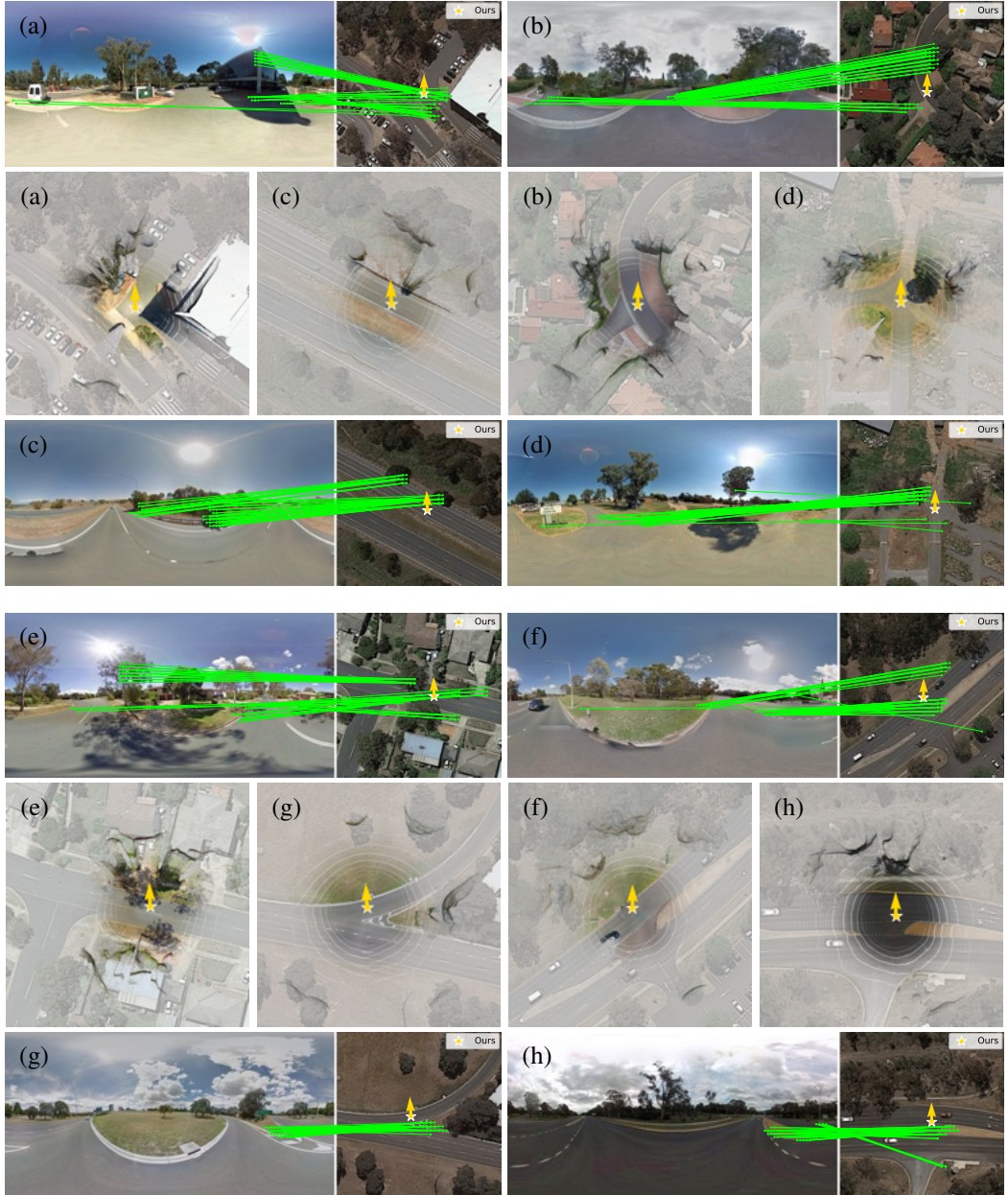

Figure 11: Direct generalization to the CVACT dataset. We visualize the top 50 correspondences, ranked by matching score, and overlay the ground layout on the aerial image after using the predicted rotation, translation, and scale transformations.

threshold is used during training with metric depth, applying the same 35 m threshold to relative depth does not select exactly the same set of objects, as our initial scale estimate only roughly maps the relative depth values to a reasonable range. As shown in Tab. 14, the impact of the maximum depth choice is minor, less than 0.2 m difference in localization errors. Since our feature matching is accurate, including or excluding relatively distant objects has little effect on pose estimation. Combined with the method's robustness to the initial scale, this makes our approach highly practical, as users do not need to carefully select an initial scale or a depth cutoff for accurate pose estimation.

**Ground-aerial layout alignment:** Next, we present additional results on ground–aerial layout alignment. We visualize the bird's-eye view ground layout after applying our estimated pose and

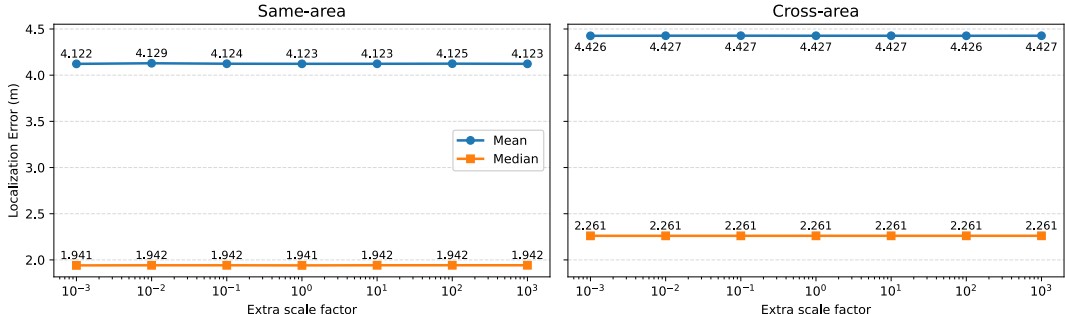

Figure 12: VIGOR test errors when different scaling factors are applied to the relative depth during inference. We use the model trained with metric depth from UniK3D (Piccinelli et al., 2025), with unknown orientation. At inference time, we use relative depth from BiFuse++ (Wang et al., 2022).

Table 14: Robustness to different maximum depth thresholds. We apply different maximum depth thresholds during inference to remove sky and distant objects (after applying the initial scale to the relative depth from BiFuse++ (Wang et al., 2022)). Best results are shown in bold. During training, we use metric depth predictions from UniK3D(Piccinelli et al., 2025), with a maximum depth threshold of 35 m.

| | Same-area | | | | Cross-area | | | |
| Max depth | ↓ Localization (m) | | ↓ Orientation (°) | | ↓ Localization (m) | | ↓ Orientation (°) | |
| | Mean | Median | Mean | Median | Mean | Median | Mean | Median |
|---|---|---|---|---|---|---|---|---|
| 25 m | **4.04** | **1.90** | **9.91** | **2.27** | **4.39** | **2.23** | **12.17** | **2.46** |
| 45 m | 4.20 | 1.97 | 10.46 | 2.35 | 4.47 | 2.27 | 12.30 | 2.51 |
| 35 m (default) | 4.12 | 1.94 | 10.13 | 2.31 | 4.43 | 2.26 | 12.27 | 2.47 |

scale. As shown in Fig. 13, the quality of alignment directly reflects localization accuracy: when the alignment is precise, the estimated pose is accurate; when the alignment is off, as in the last example, localization also fails. This interpretability provides a practical means of identifying potentially erroneous predictions.

## F  TRAINING AND TESTING WITH RELATIVE DEPTH

As mentioned in Sec. 4.3, we also experimented with training and testing using only relative depth. This setup is more challenging, since inducing correct correspondences on the BEV plane requires both ground-truth pose and scale (i.e., metric depth). As a result, the infoNCE loss cannot be directly applied and we lack direct supervision of the correspondences during training. We then explored two settings. (1) Training without infoNCE losses, i.e., setting the weight $\beta = 0$. (2) With pseudo ground truth for infoNCE losses: At each training iteration, we estimate the scale and use it to generate pseudo ground-truth positive matches for the infoNCE losses.

As expected, using relative depth during training leads to worse performance than using metric depth (see Tab. 15). When the orientation is known, finding correct correspondences between ground and aerial images is easier. In this setting, the estimated scale may quickly converge to a reasonable range, and using it to supervise correspondences ($\beta = 1$) results in better performance compared to not using it ($\beta = 0$). In contrast, when the orientation is unknown, identifying correct matches becomes significantly more difficult, making it hard to recover the correct scale. In this case, incorporating the infoNCE loss ($\beta = 1$) can introduce conflicting gradients to the pose supervision: correspondences derived from an inaccurate scale may reinforce incorrect matches, thereby degrading pose estimation performance.

As shown in Fig. 14, without explicit supervision on the correspondences ($\beta = 0$), our model still learns to establish accurate local feature matches across views. It demonstrates strong semantic understanding. For example, in Fig. 14(d), different points on the streetlight in the ground view are correctly matched to the corresponding BEV of the streetlight in the aerial view.

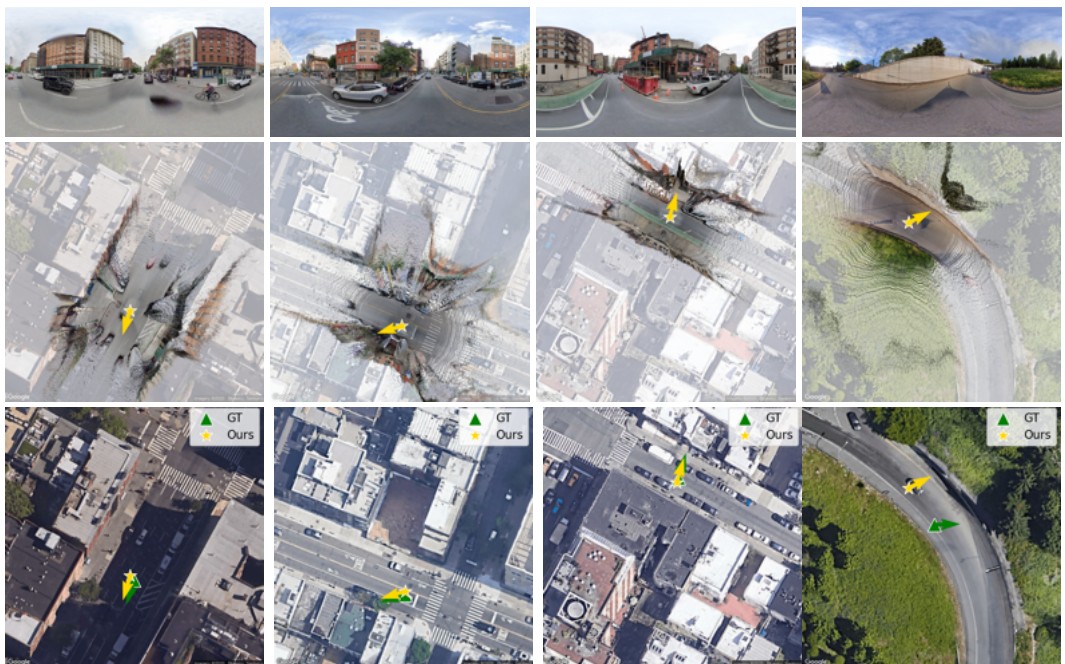

Figure 13: Ground layout overlaid on the aerial image after applying the predicted rotation, translation, and scale transformations. The alignment directly reflects localization quality: the first three examples show successful localization, while the last one illustrates a failure case. Relative depth is obtained from BiFuse++ (Wang et al., 2022).

Table 15: VIGOR test results. **Best in bold.** (1) and (2): Training and testing with relative depth predicted by BiFuse++(Wang et al., 2022). For reference, we also include the model trained and tested with metric depth from Unik3D(Piccinelli et al., 2025), as well as the model trained with metric depth and tested with relative depth.

| Ori. | Settings | Same-area | | | | Cross-area | | | |
|---|---|---|---|---|---|---|---|---|---|
| | | ↓ Localization (m) | | ↓ Orientation (°) | | ↓ Localization (m) | | ↓ Orientation (°) | |
| | | Mean | Median | Mean | Median | Mean | Median | Mean | Median |
| Known | (1) $\beta = 0$ | 3.85 | 2.33 | - | - | 4.37 | 2.74 | - | - |
| | (2) $\beta = 1$ | 3.34 | 1.99 | - | - | 3.92 | 2.39 | - | - |
| | Metric | **3.06** | **1.59** | - | - | **3.43** | **1.90** | - | - |
| | Metric + rel. | 3.17 | 1.69 | - | - | 3.56 | 2.02 | - | - |
| Unknown | (1) $\beta = 0$ | 5.86 | 3.83 | 23.09 | 9.29 | 6.72 | 4.57 | 29.44 | 12.77 |
| | (2) $\beta = 1$ | 6.25 | 4.31 | 33.40 | 16.05 | 7.28 | 4.96 | 41.05 | 20.44 |
| | Metric | **3.94** | **1.78** | **9.54** | **2.00** | **4.23** | **2.09** | **11.67** | **2.21** |
| | Metric + rel. | 4.12 | 1.94 | 10.13 | 2.31 | 4.43 | 2.26 | 12.27 | 2.47 |

## G  ADDITIONAL PROOF FOR SCALE-AWARE PROCRUSTES ALIGNMENT

As discussed in Sec. 3.2 of the main paper, we estimate pose and scale using scale-aware Procrustes alignment (Umeyama, 1991) from local correspondences. Let $\{(\mathbf{P}_n, \mathbf{Q}_n, w_n)\}_{n=1}^N$ denote weighted correspondences between ground points $\mathbf{P}$ and aerial points $\mathbf{Q}$ with weights $w_n \geq 0$ and $\sum_n w_n = 1$. We denote the weighted centroids as $\bar{\mathbf{P}}$ and $\bar{\mathbf{Q}}$, and the centered point sets as $\tilde{\mathbf{P}} = \mathbf{P} - \bar{\mathbf{P}}$ and $\tilde{\mathbf{Q}} = \mathbf{Q} - \bar{\mathbf{Q}}$. The objective is to find the optimal rotation $\mathbf{R}$, translation $\mathbf{t}$, and scale $s$ satisfying,

$$\mathbf{Q} = s(\mathbf{R} \cdot \mathbf{P}) + \mathbf{t}. \tag{7}$$

Below we provide proofs for two key steps:

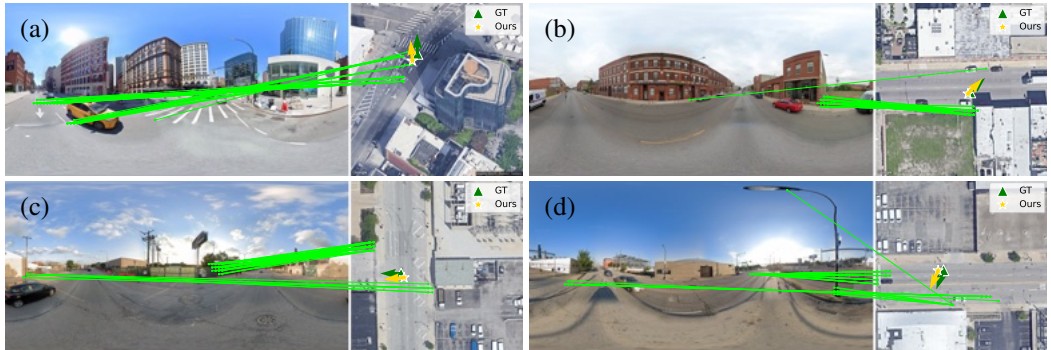

Figure 14: Local feature matching results on the VIGOR same-area test set under unknown orientation. We visualize the top 50 correspondences, ranked by matching score. The model train and test with relative depth.

(1) The optimal rotation from ground points to aerial points is $\mathbf{R} = \mathbf{V}\mathbf{U}^\top$.

(2) When ground and aerial points lie in the same metric space, i.e., there is no scale difference between them, we obtain $\mathrm{Tr}(\mathbf{\Sigma}) = \sum_{n=1}^{N} w_n \left\| \tilde{\mathbf{P}}_n \right\|^2$.

**(1) Rotation computation:** The optimal rotation between the ground and aerial points is independent of their relative scale and translation. Therefore, it is computed using only the centered point sets, $\tilde{\mathbf{Q}}$ and $\tilde{\mathbf{P}}$. We express the objective function as:

$$\mathbf{R}^* = \arg\min_{\mathbf{R}} \sum_n w_n \left\| \mathbf{R} \cdot \tilde{\mathbf{P}}_n - \tilde{\mathbf{Q}}_n \right\|^2. \tag{8}$$

The term $\left\| \mathbf{R} \cdot \tilde{\mathbf{P}}_n - \tilde{\mathbf{Q}}_n \right\|^2$ can be rewritten as (for simplicity, we omit the explicit dot product):

$$
\begin{aligned}
\left\| \mathbf{R}\tilde{\mathbf{P}}_n - \tilde{\mathbf{Q}}_n \right\|^2 &= \left( \mathbf{R}\tilde{\mathbf{P}}_n - \tilde{\mathbf{Q}}_n \right)^\top \left( \mathbf{R}\tilde{\mathbf{P}}_n - \tilde{\mathbf{Q}}_n \right) \\
&= \left( (\tilde{\mathbf{P}}_n)^\top \mathbf{R}^\top - (\tilde{\mathbf{Q}}_n)^\top \right) \left( \mathbf{R}\tilde{\mathbf{P}}_n - \tilde{\mathbf{Q}}_n \right) \\
&= (\tilde{\mathbf{P}}_n)^\top \mathbf{R}^\top \mathbf{R}\tilde{\mathbf{P}}_n - (\tilde{\mathbf{Q}}_n)^\top \mathbf{R}\tilde{\mathbf{P}}_n - (\tilde{\mathbf{P}}_n)^\top \mathbf{R}^\top \tilde{\mathbf{Q}}_n + (\tilde{\mathbf{Q}}_n)^\top \tilde{\mathbf{Q}}_n \\
&= \left\| \tilde{\mathbf{P}}_n \right\|^2 + \left\| \tilde{\mathbf{Q}}_n \right\|^2 - 2\tilde{\mathbf{Q}}_n^\top \mathbf{R}\tilde{\mathbf{P}}_n.
\end{aligned}
$$

Note that $\left\| \tilde{\mathbf{P}}_n \right\|^2$ and $\left\| \tilde{\mathbf{Q}}_n \right\|^2$ are fixed, so the optimal rotation depends only on the inner product $\tilde{\mathbf{Q}}_n^\top \mathbf{R}\tilde{\mathbf{P}}_n$. Hence, the objective in Eq. 8, which sums over all correspondences, can be written in matrix form as:

$$\mathbf{R}^* = \arg\max_{\mathbf{R}} \sum_n (w_n (\tilde{\mathbf{Q}}_n)^\top \mathbf{R}\tilde{\mathbf{P}}_n) \tag{9}$$

$$= \arg\max_{\mathbf{R}} \mathrm{Tr}(\tilde{\mathbf{Q}}^\top \mathbf{R}\tilde{\mathbf{P}}\mathbf{W}) \tag{10}$$

$$= \arg\max_{\mathbf{R}} \mathrm{Tr}(\mathbf{R}(\tilde{\mathbf{P}}\mathbf{W}\tilde{\mathbf{Q}}^\top)). \tag{11}$$

where $\mathbf{W}$ is a diagonal matrix with entrie $\mathbf{W}_{n,n} = w_n$. In Eq. 11, the matrix product $\tilde{\mathbf{P}}\mathbf{W}\tilde{\mathbf{Q}}^\top$ constructs a $2 \times 2$ square matrix $\mathbf{C}$, which can be decomposed using singular value decomposition

(SVD) as $\mathbf{C} = \mathbf{U}\boldsymbol{\Sigma}\mathbf{V}^\top$. Accordingly, the objective function in Eq. 11 can be reformulated as:

$$\mathbf{R}^* = \arg\max_{\mathbf{R}} \mathrm{Tr}(\mathbf{R}(\mathbf{U}\boldsymbol{\Sigma}\mathbf{V}^\top)) \tag{12}$$

$$= \arg\max_{\mathbf{R}} \mathrm{Tr}(\boldsymbol{\Sigma}\mathbf{V}^\top\mathbf{R}\mathbf{U}). \tag{13}$$

Since $\boldsymbol{\Sigma}$ is a $2 \times 2$ diagonal matrix with nonnegative elements, the optimal rotation is achieved when $\mathbf{V}^\top\mathbf{R}\mathbf{U} = \mathbf{I}$, where $\mathbf{R} = \mathbf{V}\mathbf{U}^\top$.

**(2) Scale computation:** As mentioned above, the $2 \times 2$ square matrix $\mathbf{C}$ obtained from the matrix product $\tilde{\mathbf{P}}\mathbf{W}\tilde{\mathbf{Q}}^\top$ can also be formulated as:

$$\mathbf{C} = \sum_{n=1}^{N} w_n \left(\tilde{\mathbf{P}}_n\right) \left(\tilde{\mathbf{Q}}_n\right)^\top = \mathbf{U}\boldsymbol{\Sigma}\mathbf{V}^\top. \tag{14}$$

The centered target points $\tilde{\mathbf{Q}}_n$ can be obtained by rotating and scaling the centered source points $\tilde{\mathbf{P}}_n$, satisfying $\tilde{\mathbf{Q}}_n = s\mathbf{R}\tilde{\mathbf{P}}_n$. With estimated rotation $\mathbf{R} = \mathbf{V}\mathbf{U}^\top$, Eq. 14 can be formulated as

$$\sum_{n=1}^{N} w_n \left(\tilde{\mathbf{P}}_n\right) \left(s\mathbf{V}\mathbf{U}^\top\tilde{\mathbf{P}}_n\right)^\top = \mathbf{U}\boldsymbol{\Sigma}\mathbf{V}^\top, \tag{15}$$

$$s\sum_{n=1}^{N} w_n \left(\tilde{\mathbf{P}}_n\tilde{\mathbf{P}}_n^\top\mathbf{U}\mathbf{V}^\top\right) = \mathbf{U}\boldsymbol{\Sigma}\mathbf{V}^\top, \tag{16}$$

$$s\sum_{n=1}^{N} w_n \left(\mathbf{U}^\top\tilde{\mathbf{P}}_n\tilde{\mathbf{P}}_n^\top\mathbf{U}\right) = \boldsymbol{\Sigma}. \tag{17}$$

Then we take the trace of both sides of Eq. 17 as

$$s\,\mathrm{Tr}(\sum_{n=1}^{N} w_n \left(\mathbf{U}^\top\tilde{\mathbf{P}}_n\tilde{\mathbf{P}}_n^\top\mathbf{U}\right)) = \mathrm{Tr}(\boldsymbol{\Sigma}), \tag{18}$$

$$s\,\mathrm{Tr}(\sum_{n=1}^{N} w_n \left(\tilde{\mathbf{P}}_n\tilde{\mathbf{P}}_n^\top\mathbf{U}\mathbf{U}^\top\right)) = \mathrm{Tr}(\boldsymbol{\Sigma}). \tag{19}$$

The trace of $\tilde{\mathbf{P}}_n\tilde{\mathbf{P}}_n^\top$ is equivalent to the norm of $\tilde{\mathbf{P}}_n$. Thus, the scale $s$ can be computed by

$$s\sum_{n=1}^{N} w_n||\tilde{\mathbf{P}}_n||^2 = \mathrm{Tr}(\boldsymbol{\Sigma}), \tag{20}$$

$$s = \frac{\mathrm{Tr}(\boldsymbol{\Sigma})}{\sum_{n=1}^{N} w_n||\tilde{\mathbf{P}}_n||^2}. \tag{21}$$

Hence, when $s = 1$, we have $\mathrm{Tr}(\boldsymbol{\Sigma}) = \sum_{n=1}^{N} w_n \left\|\tilde{\mathbf{P}}_n\right\|^2$.

## H  ADDITIONAL DETAILS ON INFONCE LOSSES

Here, we provide details on the applied infoNCE losses, $\mathcal{L}_{\text{G2S}}$ and $\mathcal{L}_{\text{S2G}}$. These losses encourage correspondences that align with the ground-truth pose while discouraging incorrect matches.

Given the sampled $N$ corresponding ground points $\mathbf{P}$ and aerial points $\mathbf{Q}$, we compute the true aerial correspondences for $\mathbf{P}$ as $\hat{\mathbf{Q}} = s(\mathbf{R}_{\text{gt}} \cdot \mathbf{P}) + \mathbf{t}_{\text{gt}}$, the true ground correspondences for $\mathbf{Q}$ as $\hat{\mathbf{P}} = \mathbf{R}_{\text{gt}}^\top \cdot ((\mathbf{Q} - \mathbf{t}_{\text{gt}})/s)$, and $s = 1$ when we use metric depth during training. Then, we find in the pairwise matching score matrix $M$ for the scores for these correspondences, denoting as $M^{\mathbf{P},\hat{\mathbf{Q}}}$ and $M^{\mathbf{Q},\hat{\mathbf{P}}}$, which will serve as the positives in the infoNCE loss.

Assuming there are $N_{\mathbf{P}}$ valid scores in $M^{\mathbf{P},\hat{\mathbf{Q}}}$ (excluding points whose correspondences fall outside the aerial image), and denoting the $n$-th score as $M_n^{\mathbf{P},\hat{\mathbf{Q}}}$, the loss $\mathcal{L}_{\text{G2S}}$ is computed as:

$$\mathcal{L}_{\text{G2S}} = -\frac{1}{N_{\mathbf{P}}} \sum_{n=1}^{N_{\mathbf{P}}} log \frac{e^{M_n^{\mathbf{P},\hat{\mathbf{Q}}}}}{\sum e^{M_i}}, \tag{22}$$

where $M_i$ is the $i$-th column correspond to the current ground point in the pairwise matching score matrix $M$. Essentially, $\mathcal{L}_{\text{G2S}}$ encourages each sampled ground point to match the aerial point found by the ground-truth pose and scale, while discouraging matches to all other aerial points.

For $\mathcal{L}_{\text{S2G}}$, we do not simply encourage the matching score $M^{\mathbf{Q},\hat{\mathbf{P}}}$ while discouraging all other matches. This is because multiple ground points may share similar planar coordinates but differ in height (i.e., the $z$ coordinate). As described in Section 3.3 of the main paper, we define a local neighborhood and use only matching scores correspond to ground points outside this neighborhood as negatives. The loss $\mathcal{L}_{\text{S2G}}$ is then computed as:

$$\mathcal{L}_{\text{S2G}} = -\frac{1}{N_{\mathbf{Q}}} \sum_{n=1}^{N_{\mathbf{Q}}} log \frac{e^{M_n^{\mathbf{Q},\hat{\mathbf{P}}}}}{\sum e^{M_{\text{neg}}}}, \tag{23}$$

where $N_{\mathbf{Q}}$ is the number of valid scores in $M^{\mathbf{Q},\hat{\mathbf{P}}}$, and $M_{\text{neg}}$ are the negatives for each sampled aerial point.

## I  DISCUSSION OF STANDARD AND TRUE DIGITAL ORTHOPHOTO MAPS

A standard Digital Orthophoto Map (DOM) is an aerial or satellite image that has been orthorectified using a Digital Elevation Model (DEM), which typically represents only the bare-earth terrain. Unlike raw aerial photographs, a DOM has a uniform scale and corrected geometric distortions, enabling it to serve as a metrically accurate base map for overlaying additional geospatial information. The aerial imagery in datasets such as VIGOR (Zhu et al., 2021) and KITTI (Shi & Li, 2022) falls into this category, as these images provide uniform scale, they still contain building facades and occlusions due to the absence of above-ground structures in the underlying DEM.

A True Digital Orthophoto Map (TDOM), in contrast, is generated using a Digital Surface Model (DSM) that captures both the terrain and above-ground objects such as buildings and vegetation. As a result, a TDOM removes facade distortions and yields an image that is geometrically much closer to a true nadir (top-down) projection. Our formulation estimates a similarity transformation (rotation, translation, and scale) between ground-level and aerial points, implicitly assuming that the aerial image behaves like a TDOM.

Despite this approximation, our method mitigates the resulting geometric inconsistencies through two key mechanisms: (1) Deep feature representations, which encode not only local appearance but also contextual information aggregated over large receptive fields, making them less sensitive to small misalignments between aerial and ground views; and (2) Confidence-based correspondence sampling, applied during both training and inference, which ensures that pose estimation relies primarily on high-quality matches. As illustrated in the main paper Fig. 3(c), the model avoids unreliable points on building facades and instead focuses on stable cues such as road markings. Likewise, in examples (a) and (d), streetlights are matched via the pole base to their aerial footprint rather than the poles themselves, which is more susceptible to perspective distortion.

## J  RUNTIME AND MEMORY USAGE

Next, we report the runtime and memory usage of our method on VIGOR in Tab. 16.

**Runtime**: On a single H100 GPU, our method achieves 14.74 FPS without RANSAC (Fischler & Bolles, 1981) (0.03s for our model and 0.04s for the depth predictor Unik3D (Piccinelli et al., 2025)) and 0.42 FPS with RANSAC. This is faster than the previous local feature matching method FG² (Xia & Alahi, 2025), which runs at 9.26 FPS without RANSAC and 0.32 FPS with it. Notably, when prioritizing speed, omitting RANSAC results in only a small increase in localization error,

| Method | Runtime (FPS) | | Memory Usage |
| --- | --- | --- | --- |
| | No RANSAC | With RANSAC | |
| FG2 | 9.26 | 0.32 | **811.52 MB** |
| Ours | **14.74** | **0.42** | 726.74 MB (base) + 1368.81 MB (Unik3D) |
| | | | 726.74 MB (base) + 117.44 MB (UniFuse) |
| | | | 726.74 MB (base) + 200.23 MB (BiFuse++) |

Table 16: Runtime and memory usage comparison.

e.g., on VIGOR same-area test set with known orientation the mean error increases from 3.06 m to 3.29 m. Furthermore, one can use a faster depth predictor to further reduce inference time.

**Memory**: our model uses 726.74 MB, with 580.54 MB for the frozen DINOv2 (Oquab et al., 2024), while the metric depth predictor Unik3D (Piccinelli et al., 2025) uses 1368.81 MB. Notably, as shown in Tab. 2, our method is also compatible with more lightweight depth models, such as Uni-Fuse Jiang et al. (2021) and BiFuse++ Wang et al. (2022).

# K   LLM USAGE STATEMENT

We used a large language model (LLM) exclusively for text polishing, limited to minor grammar corrections, wording refinements, and improvements in readability. The LLM was not involved in the design of the method, the implementation of experiments, the analysis of results, or the generation of any scientific content. All technical ideas, experiments, and conclusions presented in this work are entirely the authors' own. The role of the LLM was purely editorial, comparable to language editing support.

