# OpenReview forum: "Loc$^{2}$: Interpretable Cross-View Localization via Depth-Lifted Local Feature Matching"
_ICLR.cc/2026/Conference — ICLR 2026 Poster_

### Official Review · Reviewer_SVL7 · 2025-10-18

**Soundness:** 3
**Presentation:** 3
**Contribution:** 2
**Rating:** 6
**Confidence:** 5

**Summary:**

The paper presents a fine-grained cross-view localization approach that leverages off-the-shelf depth estimation and integrates local feature matching with scale estimation for robust pose estimation, featuring an analytic pose head that incorporates RANSAC for outlier filtering.

**Strengths:**

1. The analytic pose head, combined with RANSAC and overlay-based alignment, offers reliable diagnostics of correspondence quality.

2. The method leverages off-the-shelf depth estimation, which is already well-established and helps improve localization performance.

**Weaknesses:**

1. The method relies on off-the-shelf depth estimation, while depth provides rich geometric information, its inclusion appears to improve localization accuracy only marginally. Furthermore, were the depth models used here trained on datasets such as KITTI?

2. The results in Table 3 seem unintuitive: ignoring scale should significantly impact matching quality, yet the performance remains close to the full method. How do the authors explain this?

3. Running RANSAC on a large number of correspondences may be computationally expensive, and could its robustness degrade when the outlier ratio is high?

4. Since the approach depends heavily on local features and depth priors, it may overfit to VIGOR-style scenes and struggle to generalize to novel or unseen environments.

**Questions:**

1. Does the rotation matrix R account for pitch and roll components?

2. The results show that the method improves localization accuracy more than orientation. Do the authors have an explanation for this? You mention that orientation often aligns with road direction. Why does the local feature matching fail to capture this information effectively?

---

> ### Author Response · Authors · 2025-11-20
> **Response to Reviewer SVL7**
>
> We thank Reviewer SVL7 for the supportive feedback. All modifications in the manuscript are marked in blue. Below, we answer your questions.
>
> ---
>
> **W1:**
> We do not think that the contribution of depth can be fairly assessed by directly comparing our method with prior approaches, as they differ in more aspects than just the use of depth information. Importantly, we emphasize that incorporating depth enables a fully interpretable pipeline. In contrast, the previous interpretable method FG², which does not use depth, fails under the unknown-orientation setting.
>
> KITTI is not used for training the depth model.
> In fact, KITTI is one of the test sets used by DepthAnythingV2. Similarly, the depth models we use for VIGOR’s panoramic images (Unik3D, Bifuse++, UniFuse) are not trained on VIGOR, and VIGOR does not provide depth annotations.
>
> ---
>
> **W2:**
> We believe that an increase in mean localization error from 3.86 m to 5.47 m represents a substantial performance drop.
>
> We visually inspected the depth predictions produced by the off-the-shelf Unik3D model on VIGOR and found that they are generally of reasonable quality. As a result, the model can still learn feature matching that is biased toward these imperfect depths without completely failing. We expect that if the depth estimates were severely inaccurate, ignoring scale would have a much stronger negative impact, likely causing the feature matching to fail entirely.
>
> ---
>
> **W3:**
> Indeed, RANSAC increases computational cost, and a detailed runtime analysis is provided in Appendix Sec. I.
> In fact, RANSAC should not maintain the same level of robustness when the outlier ratio is high. This property enables us to use the number of inliers as a signal for outlier detection. Please refer to Fig. 4 for the relationship between localization error and the number of RANSAC inliers.
>
> ---
>
> **W4:**
> Cross-area testing on VIGOR already studies the generalization to unseen environments, as the model is trained on images from New York and Seattle and needs to generalize to San Francisco and Chicago. The scene layout is significantly different across cities.
>
> Furthermore, we include an additional cross-dataset generalization experiment in which our model trained on VIGOR is directly evaluated on the CVACT dataset [1], where rural and natural scenes present a significant domain generalization challenge. Despite this, our model maintains strong performance. Please refer to the updated main paper (Sec. 4.5) and Appendix Sec. D for detailed results.
>
> [1] Liu, L. and Li, H., 2019. Lending orientation to neural networks for cross-view geo-localization. In *Proceedings of the IEEE/CVF Conference on Computer Vision and Pattern Recognition* (pp. 5624–5633).
>
> ---
>
> **Q1:**
> We do not include pitch and roll in the Procrustes alignment because the aerial point coordinates are planar. However, our solution, feature matching combined with differentiable scale-aware Procrustes alignment, is generic. In principle, it can also estimate 6-DoF pose if both point sets are in 3D.
>
> ---
>
> **Q2:**
>
> ***Localization error vs. orientation error***
>
> First, regarding the improvement in localization and orientation, it is difficult to compare these two properties in isolation, as the model ultimately solves them jointly. Points that lie close to the ego-camera are far less sensitive to orientation errors, whereas points farther away experience much larger positional deviations under the same angular error. This effect is essentially the distance-amplified angular error: orientation errors produce displacement that grows proportionally with the distance from the camera.
>
> Let us take the KITTI results (with ±10° orientation noise and unknown orientation) as an example. In the ±10° case, the orientation error is already relatively small; therefore, improving the translation yields a larger gain in point alignment. In contrast, in the unknown-orientation setting (as well as on VIGOR), our method must first reduce the orientation error to a manageable range.
>
> ***Capturing data bias***
>
> Local feature–based methods can still capture dataset biases, but this is considerably less straightforward than in global feature–based approaches.
>
> Global feature–based methods often directly regress the orientation angle from learned features. Because these features encode global aerial scene layout information, it is relatively easy for the model to learn a direct mapping from scene appearance to orientation. In contrast, our method only produces local matching scores, and the orientation is subsequently solved analytically from the sampled correspondences. As a result, the model does not directly observe the global aerial layout.

---

> ### Author Response · Authors · 2025-11-27
>
> Dear Reviewer SVL7,
>
> We would like to kindly follow up on our earlier rebuttal. We understand the considerable workload involved in the review process and sincerely appreciate the time and effort you dedicate to evaluating submissions.
>
> We would be grateful if you could take a moment to review our responses to your questions, the additional experiments on cross-dataset generalization, as well as the other revisions highlighted in blue in the updated manuscript. Any feedback or comments you may have on these points would be greatly appreciated. Thank you very much for your time and consideration.
>
> Best regards,
>
> The Authors

---

> > ### Comment · Reviewer_SVL7 · 2025-11-28
> >
> > I thank the authors for their detailed and comprehensive response.
> >
> > I have read the rebuttal and the revised manuscript. The clarification regarding the training data for the depth models and the explanation of the "distance-amplified angular error" effectively addressed my concerns.
> >
> > Given that my concerns regarding generalization and the role of depth have been addressed, I will maintain my positive score.

---

### Official Review · Reviewer_rY3A · 2025-10-27

**Soundness:** 3
**Presentation:** 3
**Contribution:** 3
**Rating:** 6
**Confidence:** 3

**Summary:**

This paper proposes a cross-view localization method that first builds matches between ground and aerial images, and then analytically computes the camera pose from these matches. The idea is reasonable and demonstrates good generalization ability.

**Strengths:**

The paper is well-written and easy to follow. The idea is clear and makes sense to me.

**Weaknesses:**

1. **Purpose of “aerial points” (Line 696)**
   - What are the “aerial points” used for? I did not find a clear explanation. Do you match ground images to **regular grids** sampled on the aerial images? If yes, please justify this design choice. My concern is that downsampling features to a grid may degrade matching accuracy.

2. **Resolution vs. Accuracy: feature maps and aerial-grid density**
   - I suspect localization accuracy depends on (i) the **feature-map resolution**, (ii) the **grid resolution** on the aerial images, and (iii) the **original image resolution**. How do you balance these factors? A deeper analysis would help clarify the sensitivity to these design choices. Please provide ablations such as:
     - Fix the aerial grid and vary the **backbone** to increase feature-map resolution.
     - Fix the feature-map resolution and vary the **aerial-grid resolution**.
     - Optionally, explore multi-scale/FPN settings to study trade-offs.

3. **Temperature parameter in Equation (1)**
   - Are the matching results sensitive to the temperature parameter \( \tau \)? Please include a sensitivity analysis (e.g., performance vs. \( \tau \)) and report the chosen value and tuning protocol.

4. **Aerial image scale (ground sampling distance, GSD)**
   - Since you directly match ground to aerial features, the **GSD** (e.g., meters per pixel) will influence representational fidelity and, hence, matching accuracy. Have you evaluated aerial images captured at **higher altitudes** (coarser GSD)? Please report performance across different GSDs/heights to quantify robustness and discuss how scale changes affect the aerial feature maps and matching.

**Questions:**

See weaknesses.

---

> ### Author Response · Authors · 2025-11-20
> **Response to Reviewer rY3A (Part 1)**
>
> We thank Reviewer rY3A for the detailed feedback. All modifications in the manuscript are marked in blue. Below, we address your concerns.
>
> ---
> **W1:** *What are the “aerial points” used for?*
>
> The aerial points are used to match against all ground points to get the matching score matrix M.
>
> *Are they a regular grid sampled on the aerial images?*
>
> Intuitively, you can indeed think of them as a regular grid sampled over the aerial image.
> In practice, these aerial points come from the aerial feature map, which, when using DINOv2, is already downsampled by a factor of 14, resulting in a 45×45 feature map. Using a 41×41 grid, therefore, does not further compromise too much on the resolution.
>
> *Justify design.*
>
> Eventually, what we do is dense matching between the ground and aerial feature maps, and matching (downsampled) feature maps is common in ground-level image matching [1].
> There is a trade-off between (maybe slightly better) performance and the increasing memory usage and computation.
>
> Another small benefit of using a fixed grid of aerial points is that we maintain a consistent number of aerial points regardless of the resolution of the input aerial image. One can imagine it as resizing the feature into a fixed size. This ensures that the subsequent steps, such as the number of sampled correspondences (N) and the matching module, do not require any adjustment when the aerial image resolution changes (please refer to our reply to W4).
>
> [1] Barroso-Laguna, A., Munukutla, S., Prisacariu, V.A. and Brachmann, E., 2024. Matching 2D images in 3D: Metric relative pose from metric correspondences. In *Proceedings of the IEEE/CVF Conference on Computer Vision and Pattern Recognition* (pp. 4852–4863).
>
> ---
>
> **W2:**
> We provide the results of the suggested ablation study below.
>
> **(i) Feature map resolution:**
> We trained a model that upsamples the aerial DINO feature map from its original 45×45 resolution to 180×180 using two stages of bilinear interpolation followed by convolutional layers. We then sample a 41×41 grid of points from this higher-resolution feature map. As shown in the row “41×41 (upsampled feature)”, it does not yield better performance.
>
> **(ii)** Next, we fix the feature map resolution and train additional models using 31×31, 36×36, 46×46, 51×51, 56×56 aerial points.
> While higher resolution provides only a modest improvement in performance (0.16 meters reduction in mean localization error), it substantially increases memory usage and training time.
> More importantly, this marginal improvement mainly comes from RANSAC (which benefits from a larger pool of correspondences), and inference *without* RANSAC actually shows that a 41×41 resolution performs slightly better.
>
> **Table 1 – Different resolution configurations.**
>
> | Number of aerial points (N) | Mean loc. (m) | Median loc. (m) | Mean ori. (°) | Median ori. (°) |
> | - | - | - | - | - |
> | 31×31 | 4.22 | 1.90 | 9.68 | 1.99 |
> | 36×36 | 4.01 | 1.82 | 9.42 | 1.91 |
> | 41×41 (default)  | 3.86 | 1.75 | 9.30 | 1.93 |
> | 41×41 (upsampled feature)  | 4.55  | 1.94 | 11.46 | 1.96 |
> | 46×46 | 3.90 | 1.72 | 9.07 | 1.72 |
> | 51×51 | 3.75 | 1.67 | 8.76 | 1.73 |
> | 56×56  | **3.70** | **1.65** | **8.75** | **1.69** |
>
> **Table 2 – Different resolution configurations (inference without RANSAC).**
>
> | Number of aerial points (N) | Mean loc. (m) | Median loc. (m) | Mean ori. (°) | Median ori. (°) |
> | - | - | - | - | - |
> | 41×41 (default) | **4.60** | **2.81** | **11.19**     | **4.04**        |
> | 51×51 | 4.70 | 2.89 | 11.38 | 4.03 |
> | 56×56 | 4.68   | 2.91  | 11.59 | 3.92  |
>
> ---
>
> **W3:**
> We did a grid search by setting (\tau = 0.5, 0.1, 0.05, 0.01), and the results confirmed our choice of the widely used temperature setting (\tau = 0.1) reported in [1, 2, 3].
>
> **Table 3 – Different values for the temperature parameter.**
>
> | Temperature (\tau) | Mean loc. (m) | Median loc. (m) | Mean ori. (°) | Median ori. (°) |
> | - | - | - | - | - |
> | 0.5 | 9.58          | 6.18            | 39.03         | 15.33           |
> | 0.1 (default)      | 3.86          | **1.75**        | 9.30          | **1.93**        |
> | 0.05  | **3.78**      | 1.80            | **9.25**      | 2.17            |
> | 0.01 | 4.58          | 2.46            | 13.70         | 4.78            |
>
> [2] Sun, J., Shen, Z., Wang, Y., Bao, H. and Zhou, X., 2021. LoFTR: Detector-free local feature matching with transformers. In *Proceedings of the IEEE/CVF Conference on Computer Vision and Pattern Recognition* (pp. 8922–8931).
>
> [3] Xia, Z. and Alahi, A., 2025. FG²: Fine-Grained Cross-View Localization by Fine-Grained Feature Matching. In *Proceedings of the Computer Vision and Pattern Recognition Conference* (pp. 6362–6372).

---

> ### Author Response · Authors · 2025-11-20
> **Response to Reviewer rY3A (Part 2)**
>
> **W4:**
> Yes, we agree that the aerial image resolution (GSD) is an important factor. To study its impact, we conducted experiments with different aerial image resolutions within the same region, evaluated on the VIGOR same-area validation set under the unknown-orientation setting, and reported the results in Appendix Sec. A.1.
>
> As shown in the table below, our model remains robust, with only a slight performance drop as the aerial image resolution decreases. This degradation mainly arises from changes in the DINOv2 features caused by downsampling at lower input resolutions.
>
> **Table 4 – Different aerial image resolution.**
>
> | Resolution        | Mean loc. (m) ↓ | Median loc. (m) ↓ | Mean ori. (°) ↓ | Median ori. (°) ↓ |
> | ----------------- | --------------- | ----------------- | --------------- | ----------------- |
> | 630×630 (default) | 3.86            | 1.75              | 9.30            | **1.93**          |
> | 574×574           | **3.77**        | **1.71**          | 9.52            | 1.95              |
> | 518×518           | 3.82            | 1.76              | **9.17**        | 1.94              |
> | 448×448           | 3.90            | 1.77              | 9.49            | 1.97              |
> | 392×392           | 3.99            | 1.84              | 10.22           | 2.13              |
> | 336×336           | 4.17            | 1.96              | 10.36           | 2.28              |

---

> ### Author Response · Authors · 2025-11-27
>
> Dear Reviewer rY3A,
>
> We would like to kindly follow up on our earlier rebuttal. We understand the considerable workload involved in the review process and sincerely appreciate the time and effort you dedicate to evaluating submissions.
>
> We would be grateful if you could take a moment to review our responses to your questions, the additional ablation studies on different resolution settings and the temperature parameter, as well as the other revisions highlighted in blue in the updated manuscript. Any feedback or comments you may have on these points would be greatly appreciated.
> Thank you very much for your time and consideration.
>
> Best regards,
>
> The Authors

---

### Official Review · Reviewer_aLNR · 2025-11-01

**Soundness:** 2
**Presentation:** 3
**Contribution:** 2
**Rating:** 2
**Confidence:** 5

**Summary:**

This paper estimates the 3-DoF pose of a ground-level image based on a reference aerial image. This task is achieved by conduct local feature matching, monocular depth predictions and scale-aware Procrustes alignment in an end-to-end framework. Experiments show state-of-the-art accuracy and strong interpretability.

**Strengths:**

1 The paper is well-written and easy to follow. The  proposed framework are presented clearly.

2 The attempt to formulate cross-view matching within a differentiable training pipeline is a commendable direction.

**Weaknesses:**

1 The authors claim the method's effectiveness, but the experimental results show that it does not achieve state-of-the-art (SOTA) performance on several key metrics.

2 The method appears to be a straightforward composition of existing modules. The core challenges of cross-view matching do not seem to be fundamentally addressed by a novel technical contribution. This relates to my questions Q2-Q4.

3 The generalization ability is a major concern. The method is evaluated on a limited data split. To better validate robustness, I strongly suggest an additional cross-dataset evaluation, for instance, ​training on KITTI and testing on VIGOR.

**Questions:**

1 In the introduction, the term "scale-aware Procrustes alignment" is used. For clarity, could the authors confirm if this refers specifically to a ​2D similarity transformation​? A more detailed explanation in the text would be helpful.

2 The method seems to assume that the ground-level query images are oriented orthogonal to gravity. Could the authors explicitly state this assumption? Its practical limitation should be discussed, as consumer-grade images (e.g., from mobile phones) with arbitrary orientations may not easily be transformed into a canonical BEV space.

3 The satellite imagery used is a Digital Orthophoto Map (DOM), which does not account for building facades, unlike a True Digital Orthophoto Map (TDOM). This creates a fundamental misalignment with BEV images, especially in urban areas with tall structures. ​How does the proposed method account for or mitigate this inherent domain discrepancy?​​

4 The accuracy of the pose (location and orientation) annotations in KITTI and VIGOR is critical, as they provide the supervision signal. ​Have the authors validated the accuracy of these annotations?​​ If there is inherent noise, how does the method ensure reliable supervision?

5 Since the method establishes correspondences, an informative ablation would be to compare the performance of using a ​homography​ versus the proposed ​scale-aware Procrustes alignment.

6 The authors mention using RANSAC. ​For which specific geometric model is RANSAC employed?​​ Is it used to estimate a homography for outlier rejection before the Procrustes alignment?

---

> ### Author Response · Authors · 2025-11-20
> **Response to Reviewer aLNR (Part 1)**
>
> We thank Reviewer aLNR for the detailed feedback. All modifications in the manuscript are marked in blue. Below, we address your concerns.
>
> ---
> **W1:** We would like to clarify that, by effectiveness, we did not claim that our method achieves state-of-the-art performance on all metrics. Rather, we emphasize its strong performance under challenging conditions, such as cross-area generalization and unknown orientation, while still maintaining very competitive performance in easier settings. This all-around capability makes our method a practical and reliable choice for real-world deployment. In contrast, previous state-of-the-art baselines, such as FG2, exhibit notable performance drops when the orientation is unknown, and none of the prior methods consistently achieves state-of-the-art results across all metrics.
>
> More importantly, we hope the performance can be considered alongside our contribution to strong interpretability, which is a central focus of this work and an aspect of cross-view localization that has been largely overlooked despite its practical importance.
>
> ---
>
> **W2:** There are many important challenges in cross-view matching, and interpretability is one of them, as it enables effective outlier detection and filtering.
> Our method is not a straightforward combination of existing modules. Learning reliable feature correspondences between ground and aerial images had remained an unsolved problem. Our approach directly tackles this challenge. Our formulation of matching directly in the image plane, rather than the commonly used BEV domain, together with the explicit treatment of metric scale, constitutes a novel design.
>
> Next, we answer Q2–Q4 individually.
>
> ---
> **Q2:** Yes, we now make the assumption of knowing the gravity direction explicit in the paper (Line 215).
>
> This assumption is widely adopted in both global feature–based methods, such as SliceMatch and CCVPE, and geometry transformation–based methods, such as HC-Net, DenseFlow, GGCVT and FG^2. Since the gravity direction can be reliably estimated from a single image [1–3], as many scene objects are inherently gravity-aligned (e.g., building facades, vertical edges, vehicles), and vertical lines provide strong cues for inferring gravity, we believe this assumption is reasonable rather than restrictive.
>
> [1] Veicht, A., Sarlin, P.E., Lindenberger, P. and Pollefeys, M., 2024, September. Geocalib: Learning single-image calibration with geometric optimization. In European Conference on Computer Vision (pp. 1–20). Cham: Springer Nature Switzerland.
>
> [2] Lee, J., Go, H., Lee, H., Cho, S., Sung, M. and Kim, J., 2021. Ctrl-c: Camera calibration transformer with line-classification. In Proceedings of the IEEE/CVF International Conference on Computer Vision (pp. 16228–16237).
>
> [3] Song, X., Kang, H., Moteki, A., Suzuki, G., Kobayashi, Y. and Tan, Z., 2024. Mscc: Multi-scale transformers for camera calibration. In Proceedings of the IEEE/CVF Winter Conference on Applications of Computer Vision (pp. 3262–3271).
>
> ---
> **Q3:** Indeed, the absence of true orthophotos is a long-standing data challenge in this field. Aerial images often include building facades, occlusions, and other perspective effects. Our method mitigates these issues primarily by: (1) The use of deep features, and (2) Sampling better-matched correspondences.
>
> (1) Deep features provide each point with not only local appearance information but also contextual cues aggregated from its neighborhood. This broad receptive field helps reduce the impact of local misalignment between aerial and ground views.
>
> (2) Correspondences sampling, applied during both training and inference, allows the model to rely only on high-confidence correspondences for pose estimation. As shown in Fig. 3(c) of the paper, the model avoids points on building facades and instead leverages consistent cues such as the bus marking. Similarly, examples (a) and (d) show that, for streetlights, the model typically matches the base of the pole to its footprint in the aerial view, rather than matching the pole itself, which is influenced more by perspective effects. When using all correspondences to estimate the pose in inference, the mean and median errors increased to 6.01 and 4.78 from 3.86 and 1.76 meters, respectively.
>
> ----
> **Q4:** We understand the reviewer’s concern. We manually inspected thousands of samples and found that the ground-truth annotations are generally of good quality. The few errors we identified were found using our method, and we do not believe they introduce any meaningful statistical bias during training. In fact, minor inaccuracies in ground truth are common in many areas of computer vision and often reflect the practical realities of real-world scenarios. Both VIGOR and KITTI have been widely used and validated in prior works.

---

> ### Author Response · Authors · 2025-11-20
> **Response to Reviewer aLNR (Part 2)**
>
> **W3:** First, cross-area testing already represents a common, practical, and challenging evaluation setting. For example, on VIGOR, the model is trained on images from New York and Seattle and must generalize to San Francisco and Chicago.
>
> Additionally, we perform a cross-dataset evaluation by directly applying the model trained on VIGOR to the CVACT dataset [4]. CVACT is a cross-view image retrieval dataset collected in Canberra, Australia. The rural and natural landscapes present a significant domain generalization challenge for cross-view matching. Notably, CVACT lacks precise localization labels [5]. Hence, we provide extensive qualitative results in the main paper (new) Sec. 4.5 and Appendix Sec. D. The overlaid ground layout on the aerial image provides an effective tool for evaluation.
>
> For completeness, we also tested applying the KITTI-trained model directly to VIGOR. However, due to the drastic change in camera models, shifting from a perspective camera to a panoramic one, where image borders are continuous and correspond to the back rather than two separate sides, the model does not generalize well. This extreme setting remains an open challenge.
>
> [4] Liu, L. and Li, H., 2019. Lending orientation to neural networks for cross-view geo-localization. In Proceedings of the IEEE/CVF Conference on Computer Vision and Pattern Recognition (pp. 5624–5633).
>
> [5] Shi, Y., Yu, X., Liu, L., Campbell, D., Koniusz, P. and Li, H., 2022. Accurate 3-DoF camera geo-localization via ground-to-satellite image matching. IEEE Transactions on Pattern Analysis and Machine Intelligence, 45(3), pp. 2682–2697.
>
> ---
> **Q1:** We aim to estimate a 2D similarity transformation (rotation, translation, and uniform scaling, without reflection). Scale-aware Procrustes alignment is an analytic solution to this similarity transformation problem given planar point correspondences.
>
> ---
> **Q5:**
> We now include this ablation study. Specifically, we use the sampled correspondences to compute the homography matrix and supervise it using translation and orientation, similar to HC-Net [6]. Despite extensive hyperparameter tuning, this variant still performs poorly, achieving mean and median localization errors of 14.46 and 13.43, and orientation errors of 90.98 and 91.17 on the VIGOR same-area validation set.
>
> Notably, a homography has eight degrees of freedom, and beyond translation and orientation, it can also represent anisotropic scaling and shearing. As a result, it discards the contour and shape information encoded in the (relative) depth prior. For the same reason, it also sacrifices interpretability: the inferred ground layout can become skewed or scaled differently along different directions to align the matched points in the aerial view.
>
> [6] Wang, X., Xu, R., Cui, Z., Wan, Z. and Zhang, Y., 2023. Fine-grained cross-view geo-localization using a correlation-aware homography estimator. Advances in Neural Information Processing Systems, 36, pp. 5301–5319.
>
> [7] Wang, S., Zhang, Y., Perincherry, A., Vora, A. and Li, H., 2023. View consistent purification for accurate cross-view localization. In Proceedings of the IEEE/CVF International Conference on Computer Vision (pp. 8197–8206).
>
> ---
> **Q6:**
> The geometric model we use is the 2D similarity transformation (rotation, translation, and uniform scaling) mentioned in the reply to Q1 (not homography). Specifically, given the estimated correspondences, we find a pose and scale (computed by Procrustes alignment) that has the most correspondence inliers.

---

> ### Author Response · Authors · 2025-11-27
>
> Dear Reviewer aLNR,
>
> We would like to kindly follow up on our earlier rebuttal. We understand the considerable workload involved in the review process and sincerely appreciate the time and effort you dedicate to evaluating submissions.
>
> We would be grateful if you could take a moment to review our responses to your questions, the additional experiments on cross-dataset generalization and the homography-based pose solver, as well as the other revisions highlighted in blue in the updated manuscript. Any feedback or comments you may have on these points would be greatly appreciated.
>
> Thank you very much for your time and consideration.
>
> Best regards,
> The Authors

---

> > ### Comment · Reviewer_aLNR · 2025-11-28
> >
> > Thank you for your detailed response, which has addressed many of my previous concerns. I will increase my score accordingly. However, a few key points still require further clarification and discussion:
> >
> > W1:​ The experimental results do not yet reach state-of-the-art performance. To better understand the limitations of the proposed method, could the authors provide examples or an analysis of typical failure cases?
> >
> > Q2:​ Regarding gravity direction estimation, in practice, high-precision measurements can be directly obtained from an IMU.
> >
> > Q3:​ There is an important technical distinction to note: a standard Digital Orthophoto Map (DOM) is not related to a BEV image by a simple 2D similarity transformation; a True Digital Orthophoto Map (TDOM) is. The authors should explicitly acknowledge and discuss this approximation in the paper.
> >
> > Q4:​ Since manual selection was involved, could the authors provide a statistical summary (e.g., a table) detailing the number of images used for training and their distribution? Furthermore, manually verifying thousands of samples is highly time-consuming. Did the authors employ any semi-automated process to facilitate this?
> >
> > Q5:​ Regarding the similarity transformation, have the authors experimented with standard robust methods, such as the RANSAC-based similarity transformation available in OpenCV? It would be valuable to report these results for comparison.

---

> ### Author Response · Authors · 2025-11-29
>
> We thank Reviewer aLNR for the reply and are pleased that our initial response addressed most of the concerns. We also appreciate the willingness to increase the score. We address the remaining questions below.
>
> ---
> **W1**:
> We conducted an additional analysis of localization errors across different cities. As shown in the table below, both localization and orientation errors exhibit a clear dependence on the city. In particular, New York shows higher errors. The occlusion from dense high-rise buildings makes cross-view matching more difficult, as shown in Fig. 5(d) in our paper. We have added several additional examples in the Appendix (Fig. 10(e)–(h)) to better illustrate these cases. Furthermore, Fig. 10(a)–(d) presents other failure cases in scenes with dense vegetation or homogeneous structures (e.g., water), where the amount of matchable cross-view information becomes very sparse.
>
> Importantly, we believe these scenarios represent common challenges for all cross-view localization methods, rather than limitations specific to our approach.
>
> | Region        | Mean loc. (m) | Median loc. (m) | Mean ori. (°) | Median ori. (°) |
> | ------------- | ------------- | --------------- | ------------- | --------------- |
> | Full test set | 3.94          | 1.78            | 9.54          | 2.00            |
> | New York      | 5.07          | 2.98            | 12.04         | 2.29            |
> | Seattle       | 3.98          | 1.72            | 10.25         | 2.45            |
> | San Francisco | 2.69          | 1.29            | 5.13          | 1.49            |
> | Chicago       | 4.07          | 1.55            | 10.86         | 2.14            |
>
>
> ---
> **Q2**:
> Yes, indeed, specialized sensors such as IMUs and accelerometers can also provide measurements of the gravity direction. We have added this clarification as a footnote on page 4 of the paper.
>
> ---
> **Q3**:
> Thank you for the suggestion. We have now added a dedicated section discussing this point. Please see Appendix Sec. I.
>
> ---
> **Q4**:
> We believe there is a misunderstanding here. We use the complete official VIGOR dataset, identical to that used by all baseline methods, and we do not apply any selection or filtering to the training, validation, or test splits.
> As mentioned in our initial reply, our manual inspection of the ground-truth locations was conducted with the assistance of our model, not for filtering data but to demonstrate that our method offers strong interpretability and can help identify potential annotation errors.
>
> ---
> **Q5**:
> Yes, we additionally tested the 2D similarity transformation using OpenCV’s RANSAC-based estimator (cv2.estimateAffinePartial2D with RANSAC enabled) and did not change its default parameters.
> The results on the VIGOR same-area validation set are very similar to those obtained with our implementation.
> This consistency is expected, as both OpenCV and our method solve the same problem, i.e., estimating a 2D similarity transformation within a RANSAC loop, so any remaining differences arise only from implementation details.
>
> | VIGOR same-area val set | Mean loc. (m) | Median loc. (m) | Mean ori. (°) | Median ori. (°) |
> | ----------------------- | ------------- | --------------- | ------------- | --------------- |
> | OpenCV                  | 3.96          | 1.79            | 9.95          | 2.35            |
> | Ours                    | 3.86          | 1.75            | 9.30          | 1.93            |

---

### Official Review · Reviewer_uio4 · 2025-11-03

**Soundness:** 3
**Presentation:** 3
**Contribution:** 3
**Rating:** 6
**Confidence:** 5

**Summary:**

Loc² proposes an interpretable fine-grained cross-view localization method to estimate the 3-DoF pose of ground-level images by matching local features directly with aerial images. Unlike BEV transformation-based or global descriptor-based prior works, it learns image-plane correspondences under weak camera pose supervision, lifts ground points to BEV via monocular depth predictions, and estimates pose/scale using differentiable scale-aware Procrustes alignment. Key contributions include: direct cross-view local feature matching avoiding BEV distortion, strong interpretability via explicit correspondences and layout overlay, compatibility with both metric/relative depth, and state-of-the-art performance on KITTI and VIGOR.

**Strengths:**

- **Strong interpretability and practicality**: Explicit local correspondences enable RANSAC outlier rejection, and scaled ground layout overlay provides intuitive localization quality cues, addressing the "black box" issue of prior methods.
- **Robust and flexible depth adaptation**: Compatible with both metric and relative depth predictors, with minimal performance degradation when using lightweight relative depth models, enhancing real-world deployment potential.

**Weaknesses:**

The paper claims that 'However, these methods offer limited interpretability, as they cannot explicitly identify which objects in the ground view correspond to those in the aerial view.'  However, there is no experiments to directly compare the advantages and disadvantages of "direct image-plane matching (used in Loc²)" and "BEV transformation followed by matching". Specifically, the paper does not construct a BEV variant of Loc² (e.g., keeping the DINOv2 feature extractor, scale-aware Procrustes alignment, and loss function unchanged, but only converting the ground-level image to BEV as input for feature matching) for comparison. Without such a comparison, even though Loc² uses depth information, it cannot quantitatively prove: (1) how much performance loss BEV transformation actually causes in ground-aerial matching; (2) whether the direct matching method truly avoids the information loss of BEV transformation (e.g., whether the height information retained by direct matching with depth contributes to better correspondence accuracy).

**Questions:**

The paper argues that BEV transformation causes ray-directional distortions and height information loss, but does not conduct a direct comparative experiment between "direct image-plane matching" and "BEV transformation + matching". Have you ever constructed a BEV variant of Loc² (e.g., converting ground-level images to BEV first, then using the same DINOv2 feature extractor, matching head, and scale-aware Procrustes alignment as the original method) and compared its performance with the original direct matching？

---

> ### Author Response · Authors · 2025-11-20
> **Response to Reviewer uio4**
>
> We sincerely thank Reviewer uio4 for the constructive feedback. We appreciate your recognition that our method offers “strong interpretability and practicality” as well as “robust and flexible depth adaptation.”  All modifications in the manuscript are marked in blue. Below, we present the results of your suggested experiment.
>
> **Setting:** We construct a bird’s-eye-view (BEV) variant of Loc² by first transforming the ground-level images into BEV space and then feeding the transformed images into the same feature extractor and matching head, followed by scale-aware Procrustes alignment using the same loss functions. We adopt the BEV transformation from HC-Net [1] with its default hyperparameters, and we additionally perform a learning rate search for this new model.
>
> **Results:** Following the ablation study setting in the main paper, the results are reported on the VIGOR same-area validation set under the unknown-orientation setting. As shown in the table below, the BEV-based variant yields substantially worse performance. The BEV transformation inevitably introduces distortions to above-ground structures. These structures often occupy a significant portion of the input image and thus influence the matching process, ultimately degrading performance. Additional qualitative feature-matching results on BEV-transformed images are provided in the updated Appendix Sec. A.2.
>
> | Method design choices | Mean loc. (m) | Median loc. (m) | Mean ori. (°) | Median ori. (°) |
> | --------------------- | ------------- | --------------- | ------------- | --------------- |
> | BEV variant           | 9.49          | 8.20            | 53.42         | 31.84           |
> | **Ours**              | **3.86**      | **1.75**        | **9.30**      | **1.93**        |
>
> [1] Xiaolong Wang, Runsen Xu, Zhuofan Cui, Zeyu Wan, and Yu Zhang. Fine-grained cross-view geolocalization using a correlation-aware homography estimator. Advances in Neural Information Processing Systems, 36, 2024.

---

> ### Author Response · Authors · 2025-11-27
>
> Dear Reviewer uio4,
>
> We would like to kindly follow up on our earlier rebuttal. We understand the considerable workload involved in the review process and sincerely appreciate the time and effort you invest in evaluating submissions.
>
> We would be grateful if you could take a moment to review the additional experiments you suggested, as well as the other revisions highlighted in blue in the updated manuscript. Any feedback or comments you may have on these points would be highly appreciated.
>
> Thank you very much for your time and consideration.
>
> Best regards,
> The Authors

---

### Author Response · Authors · 2025-11-21

We sincerely thank all reviewers for their constructive feedback. We have incorporated the suggestions and carefully revised our paper, with all modifications highlighted in blue.

Notably, we added an experiment on **cross-dataset generalization**, where our model, trained exclusively on VIGOR, achieves strong correspondence estimation and localization performance on the unseen CVACT dataset.
Additional ablation studies (the BEV variant of our method, the use of a homography solver, and various hyperparameter tuning) are included in the Appendix.

---

### Author Response · Authors · 2025-12-01
**Review and Rebuttal Summary**

Dear AC,
Thank you for your time and effort in handling our submission and the discussion with the reviewers.
To assist your evaluation, we provide a concise summary of the reviewers’ feedback, our responses, and the resulting updates to the manuscript.

---
**Strengths:**

**Interpretability:**
Compared to prior cross-view localization approaches, our method exhibits "*strong interpretability and practicality*" (Reviewer uio4 and aLNR), "*addresses the black-box issue of prior methods*" (Reviewer uio4), and provides "*reliable diagnostics of correspondence quality*" (Reviewer SVL7).

**Strong generalization and practicality:**
Our method is "*robust and flexible*" w.r.t. depth models (Reviewer uio4), and "*demonstrates good generalization ability*" (Reviewer rY3A).
Importantly, Reviewers aLNR and SVL7 suggested further evaluating generalization under extreme conditions, and our new **cross-dataset generalization** experiment demonstrates the strong generalization capability of our method.

**Strong performance:**
The overall performance is also recognized by the reviewers: "*state-of-the-art performance on KITTI and VIGOR*" (Reviewer uio4) and "*the method leverages off-the-shelf depth estimation, which helps improve localization performance*" (Reviewer SVL7).

**Novel design:**
Lastly, our core method design is acknowledged by all reviewers: "*avoiding BEV distortion*" (Reviewer uio4), "*formulate cross-view matching within a differentiable training pipeline is a commendable direction*" (Reviewer aLNR), and "*the idea is clear and makes sense to me*" (Reviewer rY3A).

---
**Weakness and responses:**

**Critiques:**
Despite the strong performance acknowledged by other reviewers, Reviewer aLNR’s main critique is that the method is not state-of-the-art (SOTA) on all metrics and suggested providing examples or an analysis of typical failure cases.

**Our response:**
Our rebuttal clarified that although our approach does not achieve SOTA across every metric, it demonstrates robust performance in challenging and practical scenarios while still maintaining highly competitive performance in standard settings.
We believe the contribution of our paper lies not solely in performance, but equally in the interpretability, practicality, and generalization capabilities (highlighted in the previous strengths section).
In addition, we have included more examples and expanded our analysis of typical failure cases in Appendix Sec. C.

Apart from this, the remaining feedback primarily consists of requests for additional ablation studies and further clarifications.

**Additional ablation study:**
We conducted all suggested experiments, including model variants of using homography-transformed inputs or a homography-based pose solver, as well as analyses of various training and testing hyperparameters. All results support our initial model design and have been incorporated into the revised manuscript. We believe these additions further strengthen the paper.
Importantly, the new **cross-dataset generalization** experiment further demonstrates the strong generalization capability of our method.

**Clarifications:**
The requests for clarification regarding assumptions, terminology, ground-truth quality, and the use of RANSAC have all been addressed, and the key explanations have been incorporated into the revised manuscript.
Since most of these clarification questions were raised by Reviewers aLNR and SVL7, and both reviewers replied that their concerns were largely resolved, we believe that clarification should no longer be an issue.

---
**Summary:**
We conducted all suggested experiments and incorporated them, along with detailed clarifications addressing the reviewers’ questions, into the revised manuscript. Notably, Reviewer aLNR, the only reviewer who initially assigned a negative score, replied that they would raise their score following our first response, and we believe our subsequent replies have fully resolved all remaining concerns.
We appreciate all reviewers for their valuable feedback, which has further strengthened our paper.
Lastly, we again thank the AC for your time and effort.

---

### Meta-Review · Area_Chair_VHQe · 2026-01-02

**Summary:**

This paper proposes an interpretable cross-view localization framework based on local feature correspondences, depth lifting, and similarity-based geometric alignment. Reviewers generally found the paper to be clearly written, technically sound, and practically motivated, with strengths in interpretability, diagnostic capability, and competitive performance under certain settings.
Despite the paper’s clarity and thoughtful rebuttal, the current formulation does not yet provide a sufficiently general or principled solution to the core challenges of cross-view localization.

However, several critical concerns remain, most notably those raised by Reviewer aLNR. First, the core geometric formulation relies on a 2D similarity transform, which appears overly restrictive for realistic cross-view localization scenarios and limits the method’s ability to handle complex urban geometry. Second, although the method implicitly assumes a BEV-like alignment, it does not adequately model or resolve the systematic misalignment caused by high-rise building facades and non-orthographic aerial imagery, an issue that is fundamental rather than incidental. While the authors provided additional experiments, ablations, and clarifications in the rebuttal, these responses primarily mitigate presentation and evaluation gaps rather than addressing the underlying modeling limitations.

**Reviewer Concerns:**

**Concerns addressed by the rebuttal:**

The rebuttal successfully addressed several clarification and completeness issues raised by the reviewers. In particular, the authors added missing ablation studies (e.g., BEV variant, homography-based pose solver, resolution and temperature sensitivity), provided a cross-dataset generalization experiment (VIGOR → CVACT), and clarified implementation details and assumptions, including the gravity-alignment assumption, the role of RANSAC, and the use of off-the-shelf depth models. Reviewers also acknowledged improved discussion of failure cases, computational cost, and empirical behavior under challenging conditions. These additions resolved many of the technical and experimental questions, and led some reviewers to improve or maintain positive scores.

**Concerns that remain outstanding:**

Despite these additions, several core concerns remain unresolved. Most notably, novelty remains limited, as the method is still perceived as a relatively straightforward composition of existing components rather than a fundamentally new geometric or learning formulation. Reviewers’ concerns about the geometric adequacy of using a 2D similarity transform persist, particularly given the inherent mismatch between ground images and standard digital orthophotos (DOM) in urban environments with tall building facades; while discussed, this issue is not fundamentally resolved by the current model. In addition, the method does not consistently achieve state-of-the-art performance across key metrics, and the empirical gains from depth appear modest, raising questions about the strength of the contribution relative to prior work. Finally, although generalization experiments were added, robustness under diverse real-world conditions and sensitivity to modeling assumptions remain concerns. Taken together, these unresolved issues regarding conceptual novelty, geometric assumptions, and strength of evidence outweigh the improvements introduced in the rebuttal and informed the reject decision.

**Reviewer Scores:**

Three reviewers provide borderline above scores and one reviewer gave a clear rejection score. After reading the concerns of the reviewers, and the rebuttal of the authors, the second reviewer and the rest reviewers' scores all become positive. Though there are some technical limitations, the overall positive feedback from the reviewers second the acceptance of this work.

---

### Decision · Program_Chairs · 2026-01-26

Accept (Poster)